



# Bioconcentration as a key driver of Hg bioaccumulation in high trophic level fish

David J. Amptmeijer[1] and Johannes Bieser[1]

[1]Matter Transport and Ecosystem Dynamics, Helmhotz-Zentrum Hereon, Geesthacht, Germany

**Correspondence:** David J. Amptmeijer (davidamptmeijer@gmail.com)

**Abstract.** The ability of monomethylmercury ($MMHg^+$) to bioaccumulate in seafood is of concern due to its neurotoxic properties. Understanding the bioaccumulation of $MMHg^+$ is challenging because the $MMHg^+$ content at higher trophic levels depends on both bioconcentration and biomagnification. Furthermore, Hg can occur in several chemical species, including $Hg^{2+}$ and $MMHg^+$, which both bioaccumulate. Although the dominant pathway for $MMHg^+$ bioaccumulation into seafood is the bioconcentration of $MMHg^+$ in primary producers and the subsequent biomagnification to higher trophic levels, other pathways can contribute to $MMHg^+$ bioaccumulation. In this study, we quantify the importance of the bioaccumulation of $Hg^{2+}$ and the bioconcentration of $MMHg^+$ in higher trophic levels in the bioaccumulation of $MMHg^+$ in high trophic level fish by running a fully coupled 1D water column Hg bioaccumulation model under 3 hydrodynamic regimes typical for the North and Baltic Seas. We find that $Hg^{2+}$ bioaccumulation does not influence the bioaccumulation of $MMHg^+$ but the bioconcentration of $MMHg^+$ plays an important role. Although direct bioconcentration accounts for $< 15\%$ of $MMHg^+$ bioaccumulation in cod, the cumulative effect of bioconcentration on all trophic levels increases the $MMHg^+$ content of cod by 28-48%. We show that up to the highest trophic level modeled (TL = 3.7), the percentage of $MMHg^+$ that originates from consumer bioconcentration increases with an average of 15% per trophic level. These results demonstrate that bioconcentration in consumers is essential to accurately model the bioaccumulation of $MMHg^+$ at higher trophic levels.




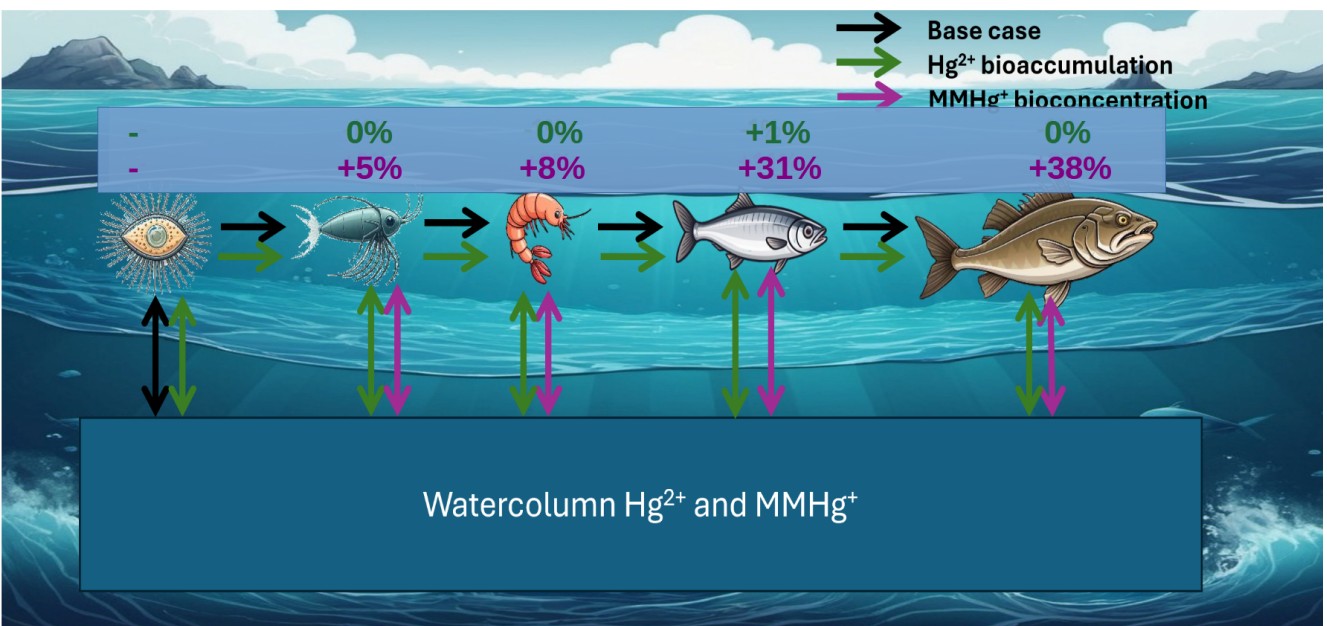

## 1  Introduction

The natural element mercury (Hg) is currently on the list of 10 substances of most concern by the World Health Organization (WHO, 2020). This is because Hg can be methylated into monomethyl mercury (MMHg$^+$). MMHg$^+$ is a topic of serious concern because MMHg$^+$ is a dangerous neurotoxin that can bioaccumulate into levels that are dangerous for human consumption in fish that are often consumed as seafood. Aquatic foods account for more than 15% of the world's protein consumption, and preserving this as a safe and reliable food source is essential to feed an increasing population (Boyd et al., 2022). Despite recent efforts, the bioaccumulation of MMHg$^+$ in the marine environment is a complex topic and is not yet fully understood. Part of the complexity of understanding MMHg$^+$ bioaccumulation and toxicity is that Hg can undergo speciation and occur in the environment in several chemical forms with distinct physical and chemical properties (Bieser et al., 2023). In particular, there are dissolved Hg (Hg$^{2+}$), dissolved elemental gaseous Hg (Hg$^0$), MMHg$^+$, and dissolved dimethylmercury (DMHg).

Often both MMHg$^+$ and DMHg are combined and are termed methylmercury (MeHg). The importance of DMHg is currently debated. Although DMHg is uncharged, which would give it higher permeability to migrate through cell membranes, it is assumed that it does not bioaccumulate to a significant degree (Morel et al., 1998). The higher bioaccumulation of MMHg$^+$ compared to DMHg can be attributed to several reasons; MMHg$^+$ can be absorbed by phytoplankton by cell-dependent factors, such as membrane channels (Garcia-Arevalo et al., 2024), volatile DMHg can more easily migrate out of cells and evaporate from the water column, and MMHg$^+$ can strongly bind to sulfhydryl (-SH) groups in organic material, notable cysteine, which traps toxic MMHg$^+$ in the cell. Although DMHg appears to be a common form of Hg in deeper water, there are no measurements in the North and Baltic Seas that would differentiate between DMHg and Hg$^0$, and its role can therefore not be assessed in the model (Fitzgerald et al., 2007). Since DMHg is susceptible to photodegradation, we can assume that it



plays an important role in the coastal water investigated in this study, until better observational studies confirm or correct this assumption (West et al., 2022) .

    Both $Hg^{2+}$ and $MMHg^+$ bioaccumulate in the marine food web. However, due to the higher toxicity and bioaccumulation potential of $MMHg^+$, the bioaccumulation of $MMHg^+$ is the most important concern and receives the most attention. There are 3 ways in which species bioaccumulate $MMHg^+$; bioconcentration, biomagnification, and in vivo methylation.

**Bioconcentration**, is the uptake of Hg directly from the water column. Because the process of bioconcentration relies on the exchange of Hg between the dissolved phase and an organism, it depends on the surface area of the organic material that is in contact with the water. Due to this, small organisms, such as bacteria and phytoplankton, have a greater ability to bioconcentrate Hg (Mason et al., 1996; Pickhardt et al., 2006). However, the bioconcentration process is complicated and recent studies show that the bioconcentration of $MMHg^+$ is influenced by cell-dependent factors, such as the thickness of the phytosphere, while this

is not the case for $Hg^{2+}$. Bioconcentration is the most important step in bioaccumulation and phytoplankton can bioconcentrate $MMHg^+$ to values between 2E4 and 6.4E6 higher than $MMHg^+$ in surrounding water (Gosnell & Mason, 2015). While the $MMHg^+$ content of phytoplankton is the most important predictor of $MMHg^+$ in higher trophic levels, it "only" predicts 63% of the variability of the $MMHg^+$ in fish (Wu et al., 2019).

    **Biomagnification** is when $MMHg^+$ reaches higher concentrations at progressively higher trophic levels. The biomagnifi-

cation factor, the fractional increase in $MMHg^+$ with each trophic level, is estimated to be $7.0 \pm 4.9$ (Harding et al., 2018; Lavoie et al., 2013). This means that in addition to the high increase in $MMHg^+$ in phytoplankton, there is a large increase in $MMHg^+$ at every consecutive trophic level. Many seafoods consist of high-trophic animals, such as cod, tuna, or marlin, which can have trophic levels between 4 and 4.8 (Nilsen et al., 2008; Sarà & Sarà, 2007). Biomagnification can increase the already high levels of $MMHg^+$ in phytoplankton by up to another factor $11.9^{4.8} \approx 145420$. The biomagnification factor of $MMHg^+$ is

extremely high, based on Lavoie et al., 2013 we estimate that the biomagnification factor for $MMHg^+$ is 1.5 times higher than for $Hg^+$. This combined with the higher toxicity of $MMHg^+$ is the reason why the bioaccumulation of $MMHg^+$ is of much higher concern than the bioaccumulation of $Hg^{2+}$.

    **In vivo methylation** occurs when animals take other forms of Hg and transform it into $MMHg^+$ in organisms. Although the existence of this process has been demonstrated in specific organisms such as cuttlefish, it is poorly understood and only

recently gaining attention (Gentè et al., 2023). There is no direct evidence of in vivo methylation in the animals that we model, so it is not implemented in this model.

    Several models model the bioaccumulation of $MMHg^+$. Notable models focus on the trophic transfer of Hg and $MMHg^+$ (Schartup et al., 2018), the bioaccumulation of $MMHg^+$ at the base of the food web (Zhang et al., 2020), the bioaccumulation of phyto and zooplankton in the Mediterranean Sea (Rosati et al., 2022) and the marine speciation and bioaccumulation in the

North and Baltic Seas by (Bieser et al., 2023), which is expanded upon in (Amptmeijer et al., 2025).

    The ECOSMO-MERCY coupled system, which is used by Bieser et al., 2023 and Amptmeijer et al., 2025 is the only coupled model that models the bioaccumulation of $Hg^{2+}$ and $MMHg^+$ at higher trophic levels such as fish while incorporating bioconcentration at every trophic level.



While MMHg$^+$ is more concerning than Hg$^{2+}$ at higher trophic levels, Hg$^{2+}$ can form up to 98% of the bioaccumulated Hg in phytoplankton (Pickhardt & Fisher, 2007). This results in a large removal of Hg$^{2+}$ during the phytoplankton bloom period (Soerensen et al., 2016). **Our first hypothesis in this study is that the bioaccumulation of Hg$^{2+}$ can lower the bioaccumulation of MMHg$^+$ by removing Hg$^{2+}$, which in turn cannot be methylated and accumulated as MMHg$^+$.**

A counterpoint is that Amptmeijer et al., 2025 does not show an average change in tHg and aqueous Hg caused by bioaccumulation, including the bioaccumulation of Hg$^{2+}$. While this indicates that there is no likely effect of Hg$^{2+}$ bioaccumulation on MMHg$^+$ bioaccumulation, it does not exclude this. Most of the bioaccumulation of Hg$^{2+}$ and MMHg$^+$ takes place at the same time and space, in the surface layer during the phytoplankton bloom. While there is no change in the average tHg, the results of Amptmeijer et al., 2025 show seasonal variation. This means that even if the average concentrations of tHg are not altered, there may still be an effect of Hg$^{2+}$ bioaccumulation on MMHg$^+$ bioaccumulation.

Most MMHg$^+$ in high trophic levels originates from their diet (Lavoie et al., 2013). So it is often assumed that the bioconcentration of MMHg$^+$ does not play an essential role in high trophic level MMHg$^+$ bioaccumulation. However, this ignores the fact that bioconcentration plays a role at all trophic levels. If both micro and mesozooplankton, for example, have 5% of MMHg$^+$ originating from bioconcentration. Mesozooplankton will have 5% less MMHg$^+$ in their diet, composed of microzooplankton, and an additional 5% less from its lack of bioconcentration. This results in a total reduction of 10%. **The second hypothesis in this paper is that the bioconcentration of MMHg$^+$ in consumers leads to a large increase in MMHg$^+$ at higher trophic levels.**

Studies that have analyzed the relative contribution of bioconcentration in the bioaccumulation of MMHg$^+$ in fish found that in freshwater fine-scale dace *Phoxinus neogaeus*, the bioconcentration accounts for up to 15% of the total bioaccumulation of MMHg$^+$ (Hall et al., 1997). A study by Wang and Wong, 2003 found that in marine sweetlips *Plectorhinchus gibbosus* bioconcentration in fish can dominate total MMHg$^+$ concentration, if they eat food with low MMHg$^+$ levels, while intake from food dominates total MMHg$^+$ uptake when fish eat food with higher MMHg$^+$ levels. This means that there will be an effect of MMHg$^+$ bioconcentration in higher trophic levels as it is a direct flux of MMHg$^+$ into the organism. In this study, we want to expand this and quantify the cumulative effect of MMHg$^+$ bioconcentration in all consumers. This allows us to evaluate if consumer bioconcentration is indeed a small percentage of total MMHg$^+$, or if it is a major contributor to the total MMHg$^+$ concentrations.

It is important to analyze these interactions using models as they cannot be fully understood using field and laboratory studies. This is because Hg$^{2+}$ and MMHg$^+$ are in active equilibrium and we cannot measure MMHg$^+$ in a system where phytoplankton would not absorb Hg$^{2+}$. The effect of bioconcentration of MMHg$^+$ can also not easily be measured. It is possible to measure the direct uptake and release of MMHg$^+$ by higher trophic levels from the water column. This is done, for example, to estimate the bioconcentration rates of Hg$^{2+}$ and MMHg$^+$ in sweetlips by Wang and Wong, 2003. The complexity is that the origin (bioconcentration or biomagnification) of MMHg$^+$ cannot be measured in observational studies, and part of the MMHg$^+$ that is consumed by higher trophic levels is bioconcentrated in consumers at the lower trophic level, making it impossible to measure the full importance of bioconcentration in consumers.





To test the 2 hypotheses that $Hg^{2+}$ bioaccumulation decreases $MMHg^+$ bioaccumulation and that $MMHg^+$ bioconcentration increases it, we quantify the effect of bioaccumulation of $Hg^{2+}$ and the bioconcentration of $MMHg^+$ on the bioaccumulation of $MMHg^+$. We do this by running the fully coupled GOTM-ECOSMO-MERCY coupled system used in Amptmeijer et al., 2025 with and without the bioaccumulation of $Hg^{2+}$ and bioconcentration of $MMHg^+$. Then we analyze the bioaccumulation of $MMHg^+$ at different trophic levels in these different scenarios and finally evaluate the importance of both interactions.

The model is run using 3 idealized 1D water column setups to represent different hydrological conditions. The setups represent the coastal hydrodynamics found in the North and Baltic Seas.

## Modeled region

The first North Sea setup is the **permanetenly mixed- southern North Sea** at $(54°15'00.0"N\ 3°34'12.0"E)$. The 41.5 m deep location of this setup is characterized by having constant water-column mixing. This remixing of nutrients within the euphoric zone creates good conditions for phytoplankton growth. Additionally, since the water column is mixed during the bloom period, macrobenthos can feed directly from the phytoplankton bloom, which results in a late population of macrobenthos. This results in a high biomass turnover rate, and macrobenthos are an important food source for predatory fish (Heip et al., 1992). The southern North Sea is rich in nutrients and the phytoplankton bloom is often light limited. Diatoms typically dominate the phytoplankton bloom in spring until silicate limitations reduce their growth and flagellates can become dominant (Emeis et al., 2015).

The second setup is the **seasonally mixed- Northern North Sea** at $(57°42'00.0"N\ 2°42'00.0"E)$. This 110 m deep setup is only seasonally mixed. The northern North Sea is still rich in nutrients resulting in similar high phytoplankton growth, which is dominated by diatoms in spring and succeded by flagellates in summer, as is the case in the southern North Sea (Bresnan et al., 2009). A key difference between the southern and northern North Sea setups is that in the northern North Sea setup macrobenthos cannot feed directly on the bloom but predominately feed on sinking detritus. This results in lower macrobenthos biomass and lower importance of macrobenthos in the diet of top predators (Heip et al., 1992).

The final set is the **permanently stratified- Gotland Deep** $(57°18'00.0"N\ 20°00'00.0"E)$. This setup is different from the 2 North Sea setups in a few ways. First, the Baltic Sea, in general, is not limited by silicate resulting in a dominance of diatoms in phytoplankton bloom. In the Gotland Deep specifically, silicate limitation can occur, but diatoms will still be more dominant. Gotland Deep has a very low salinity ( 7 g l⁻¹), is strongly stratified, and can be eutrophied in phosphate. This results in perfect growth conditions for nitrogen-fixing cyanobacteria that can form a major part of the total phytoplankton biomass in the autumn when nitrogen limitations limit the growth of other phytoplankton (Kahru & Elmgren, 2014). The presence of cyanobacteria can alter bioaccumulation because they can reduce dissolved $Hg^{2+}$ to volatile $Hg^0$, which increases Hg evaporation and therefore lowers the concentration (Kuss et al., 2015). This can reduce the average Hg content by up to 8% (Amptmeijer et al., 2025). At the same time, the small size of the cyanobacteria gives them an extremely high surface: biomass ratio, resulting in a very high bioconcentration factor of $MMHg^+$ (Pickhardt et al., 2006). Finally, the Gotland Deep has anoxic water below thermocline, because of this there is no macrobenthos (Conley et al., 2009).



The 1D setups are the same as those used in (Amptmeijer et al., 2025) and are described there in more detail. The physics of the setups is based on the Generalized Ocean Turbulence Model (GOTM) (Bolding et al., 2021), the biogeochemistry is based on the ECOSMO E2E (Daewel & Schrum, 2013), and the Hg chemistry is based on the MERCY V2.0 model (Bieser et al., 2023).

140   Quantifying the importance of the bioaccumulation of $Hg^{2+}$ and bioconcentration of $MMHg^+$ in consumers on $MMHg^+$ bioaccumulation into higher trophic levels under these idealized circumstances will provide a unique insight into the drivers of the bioaccumulation of $MMHg^+$ bioaccumulation and increase our fundamental understanding of this process. Additionally, it is important to quantify the importance of these interactions using lighter models because their inclusion in models comes at a cost. Especially the implementation of the bioaccumulation of $Hg^{2+}$ is done by adding 1 state variable to the model per

145   biota functional group, or 2 state variables if the biomagnified and bioconcentrated $Hg^{2+}$ is treated as separate variables, as is done in the model used in this study. While this is feasible without much concern in the 1D water column models that we use in this study, when running an earth system models, adding insignificant state variables becomes an unnecessary waste of computational resources which results in the wasteful expenditure of research funds and energy.

## 2   Methodolgy

150   To quantify the importance of $Hg^{2+}$ bioaccumulation and $MMHg^+$ bioconcentration on the bioaccumulation of $MMHg^+$ we modeled the bioaccumulation of $MMHg^+$ in three different scenarios using three idealized 1D water column models representing different hydrodynamic regimes typical for the North and Baltic Seas.

### 2.1   Model

Hypotheses are evaluated using the Generalized Ocean Turbulence Model (GOTM) (Bolding et al., 2021) that is coupled to the

155   ECOSMO E2E ecosystem model (Daewel et al., 2019) and the MERCY v2.0 mercury speciation model (Bieser et al., 2023). The models are coupled using the Framework for Aquatic Biogeochemical modeling (FABM) (Bruggeman & Bolding, 2014). This setup is chosen because it has been used and evaluated in previous studies to analyze the bioaccumulation and cycling of Hg in the North and Baltic Seas and it is the only fully coupled model to incorporate the bioaccumulation of $Hg^{2+}$ and the bioconcentration of $MMHg^+$ at higher trophic levels.

### 160   2.1.1   GOTM

GOTM is used to simulate the hydrodynamics of the 1D water column models. GOTM is a 1D turbulence model that computes the 1D version of the transport equation of temperature, momentum, and salinity. It does this while being nudged to observational datasets. GOTM simulations are designed using the iGOTM tool (https://igotm.bolding-bruggeman.com/). This tool compiles the observational datasets used for the GOTM simulation and estimates the water depth based on the gridded

165   bathymetry data (1/240° resolution) (GEBCO Bathymetric Compilation Group, 2020), the ECMWF ERA5 data for meteorological data, the TPOX-9 atlas for tides (1/30° resolution)(Egbert & Erofeeva, 2002), and for salinity and temperature, it uses





the wold ocean Atlas (0.25° resolution) (Garcia H.E. et al., 2019). The state is solved every 60 seconds using forward Euler differential equations. The setups have 1 grid cell per meter and the variables are exported as daily means for the post-processing analyses.

### 2.1.2 ECOSMO E2E

The ecosystem model used in this study is the ECOSMO E2E ecosystem model. The ECOSMO E2E ecosystem model is an intermediately complex ecosystem model that uses a functional group approach to estimate the biomass and carbon fluxes in the North and Baltic Seas. The version used here has 3 functional groups of phytoplankton; diatoms, flagellates, and cyanobacteria, 2 functional groups of zooplankton; microzooplankton, and mesozooplankton, 2 functional groups of fish, and 1 group of macrobenthos. The basic version of the model is published by Daewel et al., 2019, but the version used here has some modifications to make it more suitable for bioaccumulation. The modification included reducing carbon uptake efficiencies at higher trophic levels and the addition of 1 more fish functional group. This is described in detail in (Amptmeijer et al., 2025).

### 2.1.3 MERCY v2.0

The MERCY V2.0 model links atmospheric Hg to $MMHg^+$ in fish. It does this by estimating air-sea exchange and wet deposition of Hg based on the CMAQ-Hg model and calculating the marine cycling while taking into account marine speciation and bioaccumulation. It uses 35 state variables to estimate Hg speciation, transport, and bioaccumulation. Estimates the partitioning of $Hg^{2+}$ and $MMHg^+$ into dissolved organic carbon and particulate and bioaccumulation based on ecosystem parameters derived from the ecosystem model ECOSMO E2E (Bieser et al., 2023).

## 2.2 Bioaccumulation in the model

The bioaccumulation of $Hg^{2+}$ and $MMHg^+$ is the same as in (Amptmeijer et al., 2025). All functional groups of the biota can bioconcentrate both $Hg^{2+}$ and $MMHg^+$ by default. This is implemented in the form of an active equilibrium in which the bioconcentrated $Hg^{2+}$ or $MMHg^+$ on each interaction is changed by the uptake rate $m^3$ $mgC$ $d^{-1}$ multiplied by the biomass in mgC of the functional group and the concentration of Hg in ng Hg $^{-3}$. The release of Hg from the functional group is based on the respiration, mortality, and Hg release rate in $d^{-1}$. In addition to the bioconcentration, all consumer functional groups can take up Hg from the consumption of contaminated food. The uptake of $Hg^{2+}$ or $MMHg^+$ from food depends on the assimilation efficiency of the food and Hg species. After Hg has been assimilated from food $MMHg^+$ is released based on the mortality and respiration rate within the functional group while there is an additional release rate for $Hg^+$. Since fish have a temperature-dependent respiration rate in the ECOSMO E2E model, this means that fish lose Hg from both bioconcentration and biomagnification faster in warmer water as their respiration, and thus Hg release rate increases with temperature. The bioconcentration rates for zooplankton are based on Tsui and Wang, 2004, and those for fish are based on Wang and Wong, 2003.



### 2.2.1 Post-processing analysis

The post-processing analysis analysis is performed in R v4.4.1. Plots are generated using ggplot v3.5.0. and linear regression and statistics are calculated using ggpubr v0.6.0. A Wilcoxen signed rank test is performed to test the significance of differences and similarities between treatments. This is done because we assume that the trophic level influences the difference between the scenarios, which would mean that the differences between the scenarios are not normally distributed. The results are interpreted as $p < 0.05$ means a significant difference and $p > 0.05$ does not indicate a significant difference. Additionally, a Bayesian t-test is run using the BayesFactor v0.9.12-4.7 packages in R, to assess the likelihood that the scenarios are different from or the same as the base case.

### 2.3 Scenarios

The model is run in 3 different scenarios. The "Base case", "No $Hg^{2+}$ bioaccumulation" and "No $MMHg^+$ bioconcentration". The base case scenario is the same as the base case used in (Amptmeijer et al., 2025). For the "No Hg bioaccumulation" setup all uptake rates of $Hg^{2+}$ are set to zero. For the "No $MMHg^+$ bioconcentration" scenario, all consumer bioconcentrations of $MMHg^+$ and all $Hg^{2+}$ uptake rates are set to zero.

## 3 Results and discussion

### 3.1 Bioaccumulation of $Hg^{2+}$

The effect of $Hg^{2+}$ bioaccumulation on the bioaccumulation of $MMHg^+$ is shown in Table 1. The differences are low (1-5%). This is statistically evaluated, and the results are shown in Table 2. Wilcoxen's signed rank test shows that bioaccumulation of $Hg^{2+}$ has no significant impact on the bioaccumulation of $MMHg^+$ ($p > 0.99$). Furthermore, the Bayesian t-test shows that the data is 2.9 times more likely under the null hypothesis of no effect than under the alternative hypothesis. This shows that $Hg^{2+}$ bioaccumulation does not have a significant effect on $MMHg^+$ bioaccumulation (BF=0.35).

### 3.2 Bioaccumulation of $MMHg^+$

The $MMHg^+$ bioaccumulation for all biota functional groups in the different setups and scenarios and the percentage of bioaccumulated $MMHg^+$ originating from bioconcentration are shown in Table 1. The values in red in the difference category indicate when the scenario causes a change larger than 10%. The values are based on the last 10 years of the simulation and the shallow 20m of the water column, to create an average value that we can compare between the setups.

These results show that the relative contribution of bioconcentration on the $MMHg^+$ content is low in microzooplankton (4-6%) while it is higher in mesozooplankton (5-10%) higher in fish 1 (13- 22%) while lower in fish 2 (8- 14%) and higher in macrobenthos (14- 25%). The relative contribution of direct bioconcentration on the $MMHg^+$ bioaccumulation in zooplankton, especially microzooplankton, is lower than in higher trophic levels of animals. In our model, this occurs because of the



**Gotland Deep**

| | Default | | No Hg$^{2+}$ bioaccumulation | | No MMHg$^{+}$ bioconcentration | | Trophic Level |
|---|---|---|---|---|---|---|---|
| | (ng Hg mg-1) | Bioconcentrated (%) | (ng Hg mg-1) | Difference (%) | (ng Hg mg-1) | Difference (%) | - |
| Diatom | 0.0050 | 100% | 0.0050 | 0.% | 0.0050 | -0% | 1 |
| Flagellate | 0.0096 | 100% | 0.0096 | 0% | 0.0096 | -0% | 1 |
| Cyanobacteria | 0.0152 | 100% | 0.0152 | 0% | 0.0152 | 0% | 1 |
| Microzooplankton | 0.0130 | 5% | 0.0130 | 0% | 0.0123 | -5% | 2.0 |
| Mesozooplankton | 0.0190 | 5% | 0.0191 | 0% | 0.0180 | -5% | 2.2 |
| Fish 1 | 0.0314 | **16%** | 0.0314 | 0% | 0.0250 | **-20%** | 2.6 |
| Fish 2 | 0.0647 | 8% | 0.0647 | 0% | 0.0464 | **-28%** | 3.5 |

**Southern North Sea**

| | Default | | No Hg$^{2+}$ bioaccumulation | | No MMHg$^{+}$ bioconcentration | | Trophic Level |
|---|---|---|---|---|---|---|---|
| | (ng Hg mg-1) | Bioconcentrated (%) | (ng Hg mg-1) | Difference (%) | (ng Hg mg-1) | Difference (%) | - |
| Diatom | 0.0052 | 100% | 0.0049 | -5% | 0.0048 | -6% | 1 |
| Flagellate | 0.0078 | 100% | 0.0077 | -2% | 0.0076 | -2% | 1 |
| Microzooplankton | 0.0110 | 4% | 0.0108 | -2% | 0.0103 | -6% | 2.0 |
| Mesozooplankton | 0.0139 | 6% | 0.0140 | 1% | 0.0129 | -7% | 2.5 |
| Fish 1 | 0.0474 | **13%** | 0.0472 | 0% | 0.0297 | **-37%** | 3.2 |
| Fish 2 | 0.0692 | 9% | 0.0689 | -1% | 0.0423 | **-39%** | 3.5 |
| Macrobenthos | 0.0225 | **14%** | 0.0224 | -1% | 0.0168 | **-26%** | 2.3 |

**Northern North Sea**

| | Default | | No Hg$^{2+}$ bioaccumulation | | No MMHg$^{+}$ bioconcentration | | Trophic Level |
|---|---|---|---|---|---|---|---|
| | (ng Hg mg-1) | Bioconcentrated (%) | (ng Hg mg-1) | Difference (%) | (ng Hg mg-1) | Difference (%) | - |
| Diatom | 0.0034 | 100% | 0.0034 | 1% | 0.0034 | 1% | 1 |
| Flagellate | 0.0062 | 100% | 0.0062 | 1% | 0.0062 | 1% | 1 |
| Microzooplankton | 0.0104 | 6% | 0.0105 | 2% | 0.0098 | -5% | 2.0 |
| Mesozooplankton | 0.0122 | **10%** | 0.0124 | 2% | 0.0106 | **-13%** | 2.5 |
| Fish 1 | 0.0209 | **22%** | 0.0210 | 1% | 0.0135 | **-35%** | 2.9 |
| Fish 2 | 0.0373 | **14%** | 0.0374 | 0% | 0.0194 | **-48%** | 3.7 |
| Macrobenthos | 0.0085 | **25%** | 0.0083 | -2% | 0.0054 | **-37%** | 2.3 |

**Table 1.** The bioaccumulated MMHg$^{+}$, the fraction of bioaccumulated MMHg$^{+}$ that originates from bioconcentraton, and the bioaccumulated MMHg$^{+}$ in the scenario without bioaccumulation of Hg$^{2+}$ and the bioconcentration of MMHg$^{+}$ in consumers and the difference to the default scenario.





extremely high turnover rate of zooplankton. This "grow fast, die young" approach results in less MMHg$^+$ bioconcentration with higher relative contributions due to feeding caused by the high feeding rate of zooplankton.

In longer-lived fish, we see higher contributions of bioconcentration. Although these contributions are higher, they align with the experiments of (Wang & Wong, 2003) and the observations of 15% by Hall et al., 1997. Both fish 1 and fish 2 have the same
bioconcentration and release rates, so it is in line with expectations that the relative contribution of direct bioconcentration in fish 2 is lower than in fish 1 since it gets more MMHg$^+$ from its higher trophic level diet.

There is a great difference in the importance of bioconcentration of MMHg$^+$ in macrobenthos between the Southern and Northern North Sea. This difference is especially notable in the direct bioconcentration in macrobenthos, which is 25% of the total bioaccumulated MMHg$^+$ in the Northern North Sea and only 14% in the Southern North Sea. This difference is caused
by the low intake of MMHg$^+$ from food by macrobenthos in the Northern North Sea. Since the water column is stratified during spring and summer, macrobenthos cannot directly feed on the phyto- and zooplankton bloom. Because of this, they are dependent on sinking detritus. The detritus has a lower MMHg$^+$ content than living material and consequently, the MMHg$^+$ intake in Northern North Sea macrobenthos is lower, and thus the relative importance of bioconcentration is higher.

### 3.3 Bioaccumulation of MMHg$^+$ and trophic level

The relationship between trophic level and MMHg$^+$ bioaccumulation is plotted in Fig. 1a. Since we assume biomagnification to be an exponential effect per trophic level on top of bioconcentration, the model is fitted as an exponential function with the average MMHg$^+$ content of phytoplankton as the origin.

### 3.4 Evaluation hypotheses 1; The effect of Hg$^{2+}$ bioaccumulation on MMHg$^+$ bioaccumulation

Based on the results of the statistical analysis shown in table 2 we can see that there is no significant difference (p = 0.99) caused
by Hg$^{2+}$ bioaccumulation on MMHg$^+$ bioaccumulation. Based on the Bayesian t test we estimate that the change is 1/0.35 = 2.86 times greater than the data, that is, there is a difference. Based on these results, we conclude that Hg bioaccumulation$^{2+}$ does not play a major direct role in the bioaccumulation of MMHg$^+$. However, it should be noted that bioaccumulation of Hg$^{2+}$ can still play a role in the MMHg$^+$ content in biota by in vivo methylation. However, there is no data suggestion that this is a major pathway, so based on the current state of knowledge of MMHg$^{2+}$ bioaccumulation we conclude that the first hypothesis
is incorrect and Hg$^{2+}$ does not play a role in the bioaccumulation of MMHg$^+$ in coastal food webs.

### 3.5 Evaluation hypotheses 2; The effect of MMHg$^+$ bioconcentration in consumers on MMHg$^+$ bioaccumulation

Based on the statistical results shown in table 2 we conclude that there is a significant difference between the base case and the scenario without consumer bioconcentration (P < 0.001), additional we based on the Bayesian t-test that the chance that the data are different is 6.81 times larger than that no difference exists. Based on the results, we conclude that the bioconcentration
of MMHg$^+$ in consumers is a significant contributor to the bioaccumulation of MMHg$^+$. We quantify this significant increase in the bioaccumulation of MMHg$^+$ at 15% per trophic level in our model.



|  | No Hg$^{2+}$ bioaccumulation | No MMHg$^+$ consumer bioconcentration |
|---|---|---|
| Wilcoxen signed-rank test | p > 0.99 | p<0.001 |
| Bayesian t-test | BF=0.35 | BF=6.81 |

**Table 2.** The results of the statistical test performed to evaluate the difference between the scenarios and the base case. The high p-value (p > 0.99) and below 1 Bayes Factor (BF=0.35) indicate that there is no significant difference between the base case and the scenario without Hg$^{2+}$ bioaccumulation and that the change that there is no difference is 2.86 times larger than the chance that there is a difference. The difference between the scenario without MMHg$^+$ bioconcentration is significant (p < 0.001) and the change that there is a difference is 6.81 times higher than the change that there is no difference caused by the bioconcentration of MMHg$^+$ in consumers on the bioaccumulation of MMHg$^+$.

## 3.6 Model limitations

There are some limitations to our model. First, estimates of the biomagnification factor or MMHg$^+$ range between 2-10. Our model represents the estimations of the lower end. The bioconcentration factor is probably more important in low biomagnifi-
cation food webs. Another limitation is that our model stops at trophic level 3.7. This is a high trophic level that can represent piscivorous species, but many marine species that are consumed by humans such as tuna, Great Marlin, and cod can have higher trophic levels. These higher trophic levels might be influenced even more by consumers MMHg$^+$ bioconcentration. Additionally, it must be noted that fish 2 in our model is only trophic level 3.5-3.7. Typically, large cod can have a higher trophic level of 3.7-4.2 and high trophic levels such as blue fin tuna can reach trophic levels of up to 4.8 (Nilsen et al., 2008; Sarà &
Sarà, 2007).

When the absolute concentration of MMHg$^+$ increases at higher trophic levels, the relative increase in the importance of direct bioconcentration per trophic level likely decreases. Our modeled top predator with a trophic level of 3.7 has a high trophic level for a coastal ocean, but there is a marine biota with even higher trophic levels in our model domain, such as marine mammals. Without a dedicated modeling study to simulate the diet and bioconcentration of even higher trophic levels, we
cannot simply extrapolate our finding to predict the importance of MMHg$^+$ bioconcentration in their MMHg$^+$ bioaccumulation.

Overall the most important driver of our model is the fraction of MMHg$^+$ that is bioaccumulated by bioconcentration for each trophic level, as this drives the relative importance of bioconcentration at the higher trophic levels. The contribution of bioconcentration in zooplankton of 3.97-10.07% is in line with the < 20% reported by Schartup et al., 2018, and the contribution of bioconcentration in fish between 8.14-21.82% is in line with the study by Wang and Wong, 2003.
The main uncertainty for the fraction of MMHg$^+$ that originates from bioconcentration is the parameterization of bioconcentration and biomagnification. Both the bioconcentration and biomagnification of zooplankton are based on the work of Tsui and Wang, 2004 on water fleas (*Daphia Pulex*) and for fish this is based on Wang and Wong, 2003 and their work on the Indo-Pacific species Sweatlips. Although water fleas are common in the Baltic Sea, they are not in the North Sea, and sweetlips live neither in the North nor in the Baltic Sea. This means that the most important parameters in our model are not based on the
animals they represent in our model. Although it is unfeasible to have dedicated bioaccumulation studies in every animal or





functional group, there are currently not enough studies to verify whether these rates would differ between the circumstances in our model and those in the experiment. Drivers that might influence these factors are the size of the biota, physical circumstances such as temperature and salinity, or if there is a seasonal effect related to the activity of the animals. It would greatly improve our ability to model the bioaccumulation of MMHg$^+$ if more information on these different drivers were available.

It is a deliberate choice to perform this study in 1D idealized water column models, as it allows us to get a clear overview of the driving processes and generalize our findings. In this way, we can provide a general conclusion based on the biomagnification and bioconcentration rates of the biota that are presented in laboratory and field studies. However, it poses limitations compared to real fish by omitting spatial variability. Locally variable circumstances, such as the seasonally dependent flow of Hg from rivers to the ocean, could cause the importance of bioconcentration on MMHg$^+$ bioaccumulation to be regionally

different.

Although the model can predict the importance of MMHg$^+$ bioconcentration, it can not evaluate the importance of the bioconcentration of gaseous Hg species, such as Hg$^0$ and DMHg. These gaseous Hg species are assumed not to biomagnify because they are not polar but could bioconcentrate. Because the gills of fish are optimized to facilitate the exchange of gasses between water and fish blood, these gaseous Hg species can likely bioconcentrate into organisms. However, to model and

evaluate the importance of this interaction, studies must be performed first investigating both the bioconcentration and release rates of these gaseous species and their fate in the organism. In particular, the effect of the bioconcentration of DMHg on the concentration of MeHg at higher tropic levels will be very dependent on whether DMHg stays gaseous in the organism and is excreted quickly via the gills, or whether DMHg is demethylated in MMHg$^+$ and further biomagnified in the food chain. Since DMHg concentrations are low in the north and Baltic seas, this is unlikely to play a major role in our setups, but it could

influence the importance of bioconcentration on the MMHg$^+$ content of higher trophic level fish in seas with higher DMHg concentrations, such as the open oceans and the Mediterranean sea.

## 4   Conclusion

In our paper, we used a 1D water column model to test two hypotheses. Our first hypothesis is that the bioaccumulation of MMHg$^+$ is influenced by the bioaccumulation of Hg$^{2+}$. We theorized that Hg bioaccumulation$^{2+}$ removes a significant portion

of Hg$^{2+}$ from the water column, resulting in less Hg$^{2+}$ that can be methylated in MMHg$^+$. As a result, we would expect that the bioaccumulation of Hg$^{2+}$ can reduce the bioaccumulation of MMHg$^+$. Our second hypothesis is that the bioconcentration of MMHg$^+$ in consumers is a major contributor to the bioaccumulation of MMHg$^+$ at higher trophic levels. We theorized that while the direct effect of bioconcentration in high trophic level animals is low, the cumulative effect of bioaccumulation in all trophic levels below becomes a major source of MMHg$^+$

Our results show that the bioaccumulation of MMHg$^+$ in our model with and without the bioaccumulation of Hg$^{2+}$ is the same, while this is not the case for the model with and without the bioconcentration of MMHg$^+$. We show that the bioconcentration of MMHg$^+$ in consumers becomes more important at higher trophic levels because it is an effect of the sum of all trophic levels before it. We show that while direct bioconcentration only accounts for 8-14% of MMHg$^+$ bioaccumulation





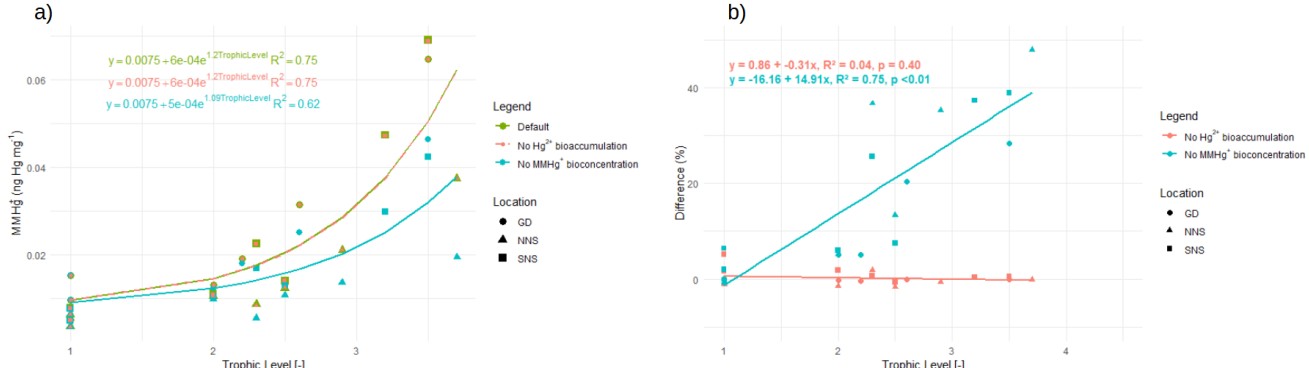

**Figure 1.** Figure a) shows the trophic level vs MMHg$^+$ bioaccumulation for the base case and the 2 scenarios. The base case and scenario "No Hg$^{2+}$ bioaccumulation" have the relationship $0.0075+6E4*e^{1.2*TrophicLevel}$ and the scenario "No MMHg$^+$ bioconcentration" has $0.0075+5E4*e^{1.09*TrophicLevel}$. Figure b) shows that there is an exponential increase in MMHg$^+$ with trophic level, which is higher for the base case and the scenario without Hg bioaccumulation than for the scenario without MMHg$^+$ bioconcentration. Figure 1 b) expands on this and demonstrates no significant effect on the importance of trophic level on the effect of the bioaccumulation of Hg$^{2+}$ on MMHg$^+$ bioaccumulation. There is a reduction of 15% per trophic level caused by the bioconcentration of MMHg$^+$.

in our highest trophic level fish, the total effect of bioconcentration in consumers accounts for 28-48%. This effect increases with the trophic level and the percentile contribution of the cumulative effect of MMHg$^+$ biooconcentration in consumers on MMHg$^+$ bioaccumulation is 15% per trophic level.

Because of this, we reject the first hypothesis that bioaccumulation of Hg$^{2+}$ lowers MMHg$^+$ bioaccumulation and accept our second hypothesis that bioconcentration of MMHg$^+$ increases bioaccumulation of MMHg$^+$ in higher trophic levels fish. We supplement the second hypothesis by quantifying the effect as an average increase in bioaccumulated MMHg$^+$ of 15% per trophic level.

These results demonstrate that to model the bioaccumulation of MMHg$^+$, the bioaccumulation of Hg$^{2+}$ can be ignored to save computational resources. However, the bioconcentration of MMHg$^+$ on the other hand is an essential interaction that should be taken into account. When modeling the bioaccumulation of MMHg$^+$ at higher trophic levels.

## Funding

This research has been funded by the Horizon 2020 research and innovation program of the European Union under the Marie Sklodowska-Curie grant agreement no. 860497.

*Author contributions.* The work was performed by David J. Amptmeijer under the supervision of Dr. Johannes Bieser.



*Competing interests.* None of the authors declares any competing interest.

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
