# Peer review of "Bioconcentration as a key driver of Hg bioaccumulation in high-trophic-level fish"

_EGUsphere, 2025_

## Author Comment (AC1)

**1 Answers to revieuwer 1**

**Reviewer Comment**

"Line 40-50. The authors mentioned the "bioconcentration", "bioaccumulation" and "bio-magnification". For instance, the statement "bioconcentration is the most important step in bioaccumulation" lacks a clear distinction from biomagnification, risking confusion for readers unfamiliar with the terminology. What's the differences between bioaccumulation and biomagnification? In the subsequent manuscript, these two words were also used in confusion. The authors should explain and clarify them. "

**Author Response**

I agree that this distinction should be clarified better, especially since it is essential for understanding the paper. I would suggest that I expand the introduction by roughly half a page in which I explain in detail the differences between bioaccumulation, biomagnification and bioconcentration and give the equations that are commonly used for the bioaccumulation, biomagnification and bioconcentration factors. I would suggest to add the replace the section of line 40-62 by the updated text below:

**Suggested edit**

**Used terminology: bioaccumulation, bioconcentration, biomagnification, and *in vivo* Hg methylation**

**Bioaccumulation** in the marine environment refers to the total increase in pollutants in biota compared to that in the water. This can be quantified in nature by measuring the concentration of pollutants in both water and biota and estimating the difference. This is typically expressed as the bioaccumulation factor, BAF. For example, the bioaccumulation of Hg in organisms $i$ can be calculated based on observations as:

$$\text{BAF}_i^{Hg} = \frac{C_i^{Hg}}{C_w^{Hg}} \tag{1}$$

In which,

$$\text{BAF}_i^{\text{Hg}} = \text{The bioaccumulation factor of MMHg for organism } i \, [\text{L} \cdot \text{kg}^{-1}]$$
$$C_i^{\text{Hg}} = \text{The concentration of MMHg}^+ \text{ in organism } i \, [\text{ng Hg} \cdot \text{kg}^{-1}]$$
$$C_w^{\text{Hg}} = \text{The free concentration of Hg}^+ \text{ in water} \, [\text{ng Hg} \cdot \text{L}^{-1}]$$

Where Hg could either refer to $Hg^{2+}$ or $MMHg^+$. Since the BAF can be based on field measurements, it is a commonly used metric to estimate the link between the concentrations of pollutants in seawater and those in biota. In this study, we are interested in separating the bioaccumulation into separate pathways: the direct uptake from the water (bioconcentration) and the increase in pollutants due to trophic interactions (biomagnification).

**Bioconcentration**, is the increase in the concentration of Hg in biota directly due to uptake from the water. Because the process of bioconcentration relies on the exchange of Hg between the dissolved phase and an organism, it depends on the surface area of the organic material that is in contact with the water. Due to this, small organisms, such as bacteria and phytoplankton, have a greater ability to bioconcentrate Hg (Mason et al., 1996; Pickhardt et al., 2006). However, the bioconcentration process is complicated and recent studies show that the bioconcentration of $MMHg^+$ is influenced by cell-dependent factors, such as the thickness of the phycosphere and the availability of transmembrane

channels, while this is not the case for $Hg^{2+}$ (Garcia-Arevalo et al., 2024). Bioconcentration is typically defined by the bioconcentration factor (BCF). The BCF for $MMHg^+$ in organism $i$, can for example be calculated as

$$\text{BCF}_i^{Hg} = \frac{BC_i^{Hg}}{C_w^{Hg}} \tag{2}$$

In which,

$$\text{BCF}_i^{\text{Hg}} = \text{The bioconcentration factor of Hg for organism } i \, [\text{L} \cdot \text{kg}^{-1}]$$
$$BC_i^{\text{Hg}} = \text{The concentration of Hg in organism } i \text{ due to direct bioconcentration} \, [\text{ng Hg} \cdot \text{kg}^{-1}]$$
$$C_w^{\text{Hg}} = \text{The free concentration of Hg in the water} \, [\text{ng Hg} \cdot \text{L}^{-1}]$$

Here, Hg could either refer to $Hg^{2+}$ or $MMHg^+$. Note that this defines the theoretical BCF. In nature it is typically impossible to directly measure the BCF, as it would be impossible to separate between $MMHg^+$ that is taken up directly from the water and $MMHg^+$ that is ingested via food. Bioconcentration is the most important step in bioaccumulation and phytoplankton can have a BCF of $MMHg^+$ between 2E4 L kg$^{-1}$ and 6.4E6 L kg$^{-1}$ (Gosnell & Mason, 2015).

**Biomagnification** is when $MMHg^+$ reaches higher concentrations at progressively higher trophic levels. The biomagnification factor, the fractional increase in $MMHg^+$ with each trophic level, is estimated to be $7.0 \pm 4.9$ (Harding et al., 2018; Lavoie et al., 2013). This means that in addition to the high increase in $MMHg^+$ in phytoplankton, there is a large increase in $MMHg^+$ at every consecutive trophic level. Many seafoods consist of high-trophic animals, such as cod, tuna, or marlin, which can have trophic levels between 4 and 4.8 (Nilsen et al., 2008; Sarà & Sarà, 2007). Biomagnification can increase the already high levels of $MMHg^+$ in phytoplankton by up to another factor $11.9^{4.8} \approx 145420$. This is typically defined by the biomagnification factor, BMF, which can be calculated assuming steady state for organism i, preying on organism j for $MMHg^+$ as:

$$\text{BMF}_{i,j}^{Hg} = \frac{C_i^{Hg}}{C_j^{Hg}} \tag{3}$$

In which,

$$\text{BMF}_{ij}^{\text{Hg}} = \text{The biomagnification factor for trophic consumption of organism } j \text{ by } i \, [\text{unitless}]$$
$$C_j^{\text{Hg}} = \text{The concentration of Hg in organism } j \, [\text{ng Hg} \cdot \text{kg}^{-1}]$$
$$C_i^{\text{Hg}} = \text{The concentration of Hg in organism } i \, [\text{ng Hg} \cdot \text{kg}^{-1}]$$

The biomagnification factor of $MMHg^+$ is extremely high, Lavoie et al., 2013 estimates the diet-weighted average BMF for $MMHg^+$ as $8.1 \pm 7.2$ while it is only $4.7 \pm 4.7$ for $Hg^{2+}$. This combined with the higher toxicity of $MMHg^+$ is the reason why the bioaccumulation of $MMHg^+$ is of much higher concern than the bioaccumulation of $Hg^{2+}$.

**In vivo methylation** occurs when animals take other forms of Hg and transform it into $MMHg^+$ in organisms. Although the existence of this process has been demonstrated in specific organisms such as cuttlefish, it is poorly understood and only recently gaining attention (Gente et al., 2023). There is no direct evidence of in vivo methylation in the animals that we model, so it is not implemented in this model.

Overall the dominant pathway of the bioaccumulation of $MMHg^+$ is the bioconcentration of $MMHg^+$ in phytoplankton and consequent biomagnification. The important route is

quantified by Wu et al., 2019 using a meta-analysis. They find that the concentration of MeHg at the base of the food web predicts 63% of the observed variability in high trophic level fish, while the remaining 37% is controlled by factors such as the dissolved organic matter content and oligotrophy.

**Author Response**

Than I would suggest to expand section 2.2 in the methods section (line 184) to explain the exact equations used in this paper to asses bioconcentration and bioaccumulation.

**Suggested edit**

The implementation of bioaccumulation is discussed and validated in more detail in Amptmeijer et al., 2025, but the core equations are discussesed here as well for clarity.
The increase in bioconcentrated pollutant ($Hg^{2+}$ or $MMHg^{+}$) per day for a functional group is calculated based on the biomass concentration of the group, the uptake rate, and the concentration of the pollutant, while it is reduced with a rate that is the sum of the release rate of the pollutant and the loss of biomass from group g, from both biologicall loss (respiration and mortality) and predation. The change in pollutant p due to bioaccumulation can then be calculated using the following equation:

$$\frac{dC_{g,p}^{BC}}{dt} = b_g \cdot C_p^{env} \cdot r_{g,p}^{bc} - C_{g,p}^{BC} \cdot (r_{g,p}^{rel} + r_g^{bl} + \sum_{z=1}^{n_z} r_{z,g}^{pred}) \tag{4}$$

$$
\begin{aligned}
C_{g,p}^{BC} &= \text{Bioconcentrated pollutant } p \text{ in group } g \text{ [ng Hg m}^{-3}\text{]} \\
b_g &= \text{Biomass of functional group } g \text{ [mgC m}^{-3}\text{]} \\
C_p^{env} &= \text{Environmental concentration of pollutant } p \text{ [ng Hg m}^{-3}\text{]} \\
r_{g,p}^{bc} &= \text{Bioconcentration rate for group } g \text{ and pollutant } p \text{ [ng Hg mgC}^{-1}\text{ d}^{-1}\text{]} \\
r_{g,p}^{rel} &= \text{Release rate of pollutant } p \text{ from group } g \text{ [d}^{-1}\text{]} \\
r_g^{bl} &= \text{Biological loss rate for group } g \text{ (mortality, respiration) [d}^{-1}\text{]} \\
r_{z,g}^{pred} &= \text{Predation rate by predator } z \text{ on group } g \text{ [d}^{-1}\text{]} \\
n_z &= \text{Number of consumer groups feeding on group } g \\
z &= \text{Index for consumer groups (predators) of } g \\
t &= \text{Time [d]}
\end{aligned}
$$

While the change in pollutant p due to biomagnfication is also dependent on the predation and concentration of pollutants from both bioconcentration and biomagniciation in the prey. Additionally pollutant p is released via the turnover rate rather than the release rate as is the case for bioconcentration, the change in pollutant p due to biomagnification can then be calculated as follows:

$$\frac{dC_{g,p}^{BM}}{dt} = \sum_{s=1}^{n_s} (r_{g,s}^{pred} \cdot a_{s,p} \cdot (C_{s,p}^{BC} + C_{s,p}^{BM})) - C_{g,p}^{BM} \cdot (r_{p,g}^{to} + r_g^{bl} \sum_{z=1}^{n_z} r_{z,g}^{pred}) \tag{5}$$

So the total concentration of pollutant P in ng Hg m$^{-3}$ is:

$$C_{(g,p)} = C_{(g,p)}^{BC} + C_{(g,p)}^{BM} \tag{6}$$

Since this tracks the pollutants per volume of water, the total bioaccumulation per biomass in ng Hg mgC$^{-1}$ is then calculated as

$$C_{(g,p)}^{bg} = \frac{C_{(g,p)}}{b_g} \qquad (7)$$

This is then converted to the bioaccumulation per dry weight based on an assumed ratio of carbon to dry weight of 0.2 for diatoms, 0.33 for flagellates and cyanobacteria, and 0.5 for zooplankton and fish based on Walve and Larsson, 1999 and Sicko-Goad et al., 1984.

**Reviewer Comment**

"Line 79-85. The second hypothesis is confused. This hypothesis lacks evidence and references, making it appears speculative."

**Author Response**

I agree that the second hypothesis lacked references. It is mostly supported by the work of Wu et al., 2019, so I would rewrite that as:

**Suggested edit**

The majority of MMHg$^+$ present in higher trophic levels is derived from their dietary intake (Lavoie et al., 2013). It is often assumed that MMHg$^+$ bioconcentration is not crucial for its bioaccumulation at higher trophic levels based on results such as those presented by Schartup et al., 2018, therefore it is, for example omitted from several Hg cycling and bioaccumulation models such as the model presentd by Rosati et al., 2022, or not incoroporated into higher trophic level as is the case in the model presented by Li et al., 2024. However, this assumption overlooks that bioconcentration occurs at all levels of the trophic hierarchy. For example, if microzooplankton and mesozooplankton acquire 5% of MMHg$^+$ through bioconcentration, mesozooplankton will have 5% less MMHg$^+$ from its diet, which consists of microzooplankton, and another 5% less due to absence of bioconcentration, leading to a total reduction of 10%. **The second hypothesis proposed here is that MMHg$^+$ bioconcentration in consumers significantly elevates MMHg$^+$ levels at higher trophic levels.** This concept has been previously suggested and studied by Wu et al., 2019. Their research found that the BCF in fish spans 3 to 7 orders of magnitude and greatly differs across studied sites, yet they did find a strong correlation between BCF and MMHg$^+$ concentration in fish. Thus, we are not the first to suggest that direct water uptake is a significant factor in MMHg$^+$ bioaccumulation, rather, this study extends this understanding by quantifying the role of bioconcentration in all consumers on MMHg$^+$ bioaccumulation in fish at higher trophic levels.

**Author Response**

Additionally, I think the first hypothesis can also be better phrased. It was added as by itself it is not a negative result that I believe is interesting enough to publish, but it could supplement this study. I would howeever propose 2 changes. I would suggest to refrase its introduction and to expand the results related to this as discussed below:

I would refrase the first hypothesis (line 72) as:

**Suggested edit**

While MMHg$^+$ is more concerning than Hg$^{2+}$ at higher trophic levels, Hg$^{2+}$ can form up to 98% of the bioaccumulated Hg in phytoplankton (Pickhardt & Fisher, 2007). This results in a large removal of Hg$^{2+}$ during the phytoplankton bloom period (Soerensen et al., 2016). However it is demonstrated by Amptmeijer et al., 2025, which analyses the feedback of bioaccumulation on Hg cycling, that there is no change in average tHg and aqueous Hg caused by bioaccumulation, but that there is a seasonal variation in the aquatic tHg content due to bioaccumulation. This means that even if the average concentrations of tHg are not altered by bioaccumulation, there may still be an effect of Hg$^{2+}$ bioaccumulation on MMHg$^+$ bioaccumulation as during the phytoplankton bloom tHg is reduced which could lead to a reduction of available MMHg$^+$ for bioaccumulation. It could be theorized that as the ecosystem reduces tHg during the phytoplankton bloom, it would reduce dissolved MMHg$^+$, as this is in active equilibrium with other Hg species and therefore reduce the availability of MMHg$^+$ for bioaccumulation. Based on this we propose our first hypothesis that **the bioaccumulation of Hg$^{2+}$ can lower the bioaccumulation of MMHg$^+$ by removing Hg$^{2+}$, which in turn cannot be methylated and accumulated as MMHg$^+$.**

**Author Response**

As the first hypothesis is focussed mostly on the theory that seasonal bioaccumulation of Hg$^{2+}$ could influence MMHg$^+$ bioaccumulation I suggest that we also expand the results by looking if there is no seasonal effect. For this I would suggest to add the following section before the hypothesis evaluation (line 249 in the preprint.)

**Suggested edit**

**Seasonality of the difference in MMHg$^+$ bioaccumulation**

The seasonality of the difference in MMHg$^+$ bioaccumulation caused the bioaccumulation of Hg$^{2+}$ and the bioconcentration of MMHg$^+$ in consumers is shown in Fig. 1. For each calendar day (January 1$^{st}$, January 2$^{nd}$, etc.), the modeled daily values from each of the last 10 years of the simulations were averaged. The resulting time series represents an annual cycle of average daily conditions. From the producers functional groups only the diatoms are shown as the reaction is not group specific but rather caused by changes in dissolved Hg$^{2+}$ and MMHg$^+$ which means the diffference caused for all phytoplankton groups was the same. This shows that, while the scale depends on the setup, there are interactions that consistently occur. In low trophic levels such as phytoplankotn and microzooplankton the bioaccumulation of Hg$^{2+}$ causes a seasonal respone in the MMHg$^+$ bioaccumulation in phytoplankton which is consequently observable in low trophic level biota such as microzooplankton. While this reduction in MMHg$^+$ would compound into higher trophic levels, its effects in higher trophic level animals dwarves in comparison to the difference caused by incorporating the bioconcentration of MMHg$^+$ in consumers and it does not cause a difference larger than 3% in either fish 1 or fish 2 in any of the setups.

**Author Response**

Then I would also suggest to update the hypotheses evaluation (section 3.3 line 238) in the result section as follows

> **Suggested edit**
>
> ### Evaluation hypotheses 1; The effect of $Hg^{2+}$ bioaccumulation on $MMHg^+$ bioaccumulation
>
> Based on the results of the statistical analysis shown in Table 2, we can see that there is no significant difference (p = 0.99) caused by $Hg^{2+}$ bioaccumulation on $MMHg^+$ bioaccumulation. Based on the Bayesian t-test, we estimate that the change is $1/0.35 = 2.86$ times greater than the data, meaning there is a difference. We do note that the seasonal changes in the total Hg concentration due to bioaccumulation change the bioaccumulation of $MMHg^+$ at the base of the food web, and we can see this change in phytoplankton and low trophic level consumers, but it does not cause a notable ($> 5\%$) change in $MMHg^+$ bioaccumulation in fish. Based on these results, we conclude that $Hg^{2+}$ bioaccumulation does not play a major direct role in the bioaccumulation of $MMHg^+$. However, it should be noted that the bioaccumulation of $Hg^{2+}$ can still play a role in the $MMHg^+$ content in biota by in vivo methylation. However, there is no data suggesting that this is a major pathway, so based on the current state of knowledge of $MMHg^+$ bioaccumulation, we conclude that the first hypothesis is incorrect and $Hg^{2+}$ does not play a role in the bioaccumulation of $MMHg^+$ in coastal food webs.

[Figure]

Figure 1: The seasonality of the difference in the bioaccumulation of $MMHg^+$ caused by the bioaccumulation of $Hg^{2+}$ and the bioconcentration of MMHg in consumers for a) the Gotland Deep, b) the Northern North Sea and c) the Southern North Sea. In high trophic level such as fish 1 and fish 2 there is low seasonality and the effect of the bioconcentration of $MMHg^+$ in consumers is high while the effect of the bioaccumulation of $Hg^+$ is low. In low trophic levels, notably diatoms and microzooplankton there is strong seasonal component. The bioaccumulation of $MMHg^+$ is up to 5% lower in diatoms in the Southern North Sea if the bioaccumulation of $Hg^{2+}$ is modeled in late summer when biomass is high. But the bioaccumulation of $Hg^{2+}$ does not lead to a notable ($> 5\%$) difference at any moment in fish.

> **Reviewer Comment**
>
> "Line 140: As mentioned, "Quantifying the importance of the bioaccumulation of Hg2+ and bioconcentration of MMHg+ in consumers on MMHg+ bioaccumulation". The authors should clarify whether prior models ignored multi-trophic bioconcentration, and then highlight the novelty of this work."

**Author Response**

Thank you for your excelent suggestions, I would suggest expanding the introduction by adding the following section to replace to short discussion on previous models at line 62. Then it would be more thoroughly revieuwed when the readers arrives at line 140.

**Suggested edit**

**Current models**

Multiple models have been developed to explain $MMHg^+$ bioaccumulation in marine ecosystems. Key examples include trophic transfer (Schartup et al., 2018), base-level accumulation (Zhang et al., 2020), planktonic bioaccumulation in the Mediterranean Sea (Rosati et al., 2022), MeHg dynamics on the Beaufort Shelf (Li et al., 2022), and speciation plus accumulation in the North and Baltic Seas (Bieser et al., 2023).

In all previous models, the bioconcentration of $MMHg^+$ is included because it is an essential driver. These models, however, do not include higher trophic level animals such as fish. It is concluded in Schartup et al., 2018 that the bioconcentration of $MMHg^+$ in zooplankton is not a major contributor and contributes less than 15% of total MeHg bioaccumulation. Consequently, in later models such as presented by Rosati et al., 2022 this interaction is not included because their model focuses on the base of the food web. The study performed by Li et al., 2022 includes the process of bioconcentration for invertebrates, but it is not included for vertebrates. This means that our model would be the first model to include bioconcentration at every trophic level.

The bioaccumulation of $Hg^{2+}$ is much less studied and not incorporated in any of the above-mentioned models. This is because $Hg^{2+}$ is much less toxic than $MMHg^+$ and therefore comparably understudied. While data is limited, this raises the speculative question if the link between the bioaccumulation $Hg^{2+}$ and $MMHg^+$ is not underestimated as $Hg^{2+}$ and $MMHg^+$ are in active equilibrium in the water.

The ECOSMO-MERCY coupled system, which is used by Bieser et al., 2023 and Amptmeijer et al., 2025 is the only coupled model that models the bioaccumulation of $Hg^{2+}$ and $MMHg^+$ at higher trophic levels such as fish, while incorporating bioconcentration at every trophic level.

**Reviewer Comment**

"Line 226 Table 1. How to calculate the bioaccumulation and bioconcentration difference (%) ?

**Author Response**

Thank you for the noticing. It is cacluated as scenario/base case * 100%, but I will add this to the paper as this specification is indeed essential. I would suggest to add to section 3.2 to following clarification:

**Suggested edit**

The % bioconcentrated is calculated as Bioconcentrated (%) $= \frac{\text{Bioconcentrated}}{\text{Bioaccumulated}}$ and the difference (%) is calculated as Difference (%) $= \frac{\text{Scenario}}{\text{Default}}$. The values in red in the difference category indicate when the scenario causes a change larger than 10%. The

values are based on the last 10 years of the simulation and the top 20m of the water column, to create an average value that we can compare between the setups.

**Author Response**

And update the caption of Table 1 to also include this as follows:

**Suggested edit**

The bioaccumulated $MMHg^+$, the percentage of bioaccumulated $MMHg^+$ that originates from bioconcentraton, and the bioaccumulated $MMHg^+$ in the scenario without bioaccumulation of $Hg^{2+}$ and the bioconcentration of $MMHg^+$ in consumers and the difference to the default scenario. The % bioconcentrated is calculated as Bioconcentrated (%) = $\frac{\text{Bioconcentrated}}{\text{Bioaccumulated}}$ and the difference (%) is calculated as Difference (%) = $\frac{\text{Scenario}}{\text{Default}}$.

**Reviewer Comment**

Line 316 "15% per trophic level". It is recommended to supplement sensitivity analyses.

**Author Response**

Thank you for this valuable suggestion. I would suggest to edd the following section to the methods section to introduce the sensitivity analyses:

**Suggested edit**

In order to further investigate how bioconcentration in consumers affects bioaccumulation of $MMHg^+$, we performed a sensitivity analysis on the key drivers: the bioconcentration rate of consumers and the bioaccumulation rate of producers. To this extent, two sensitivity studies are performed. In the first sensitivity study, the bioconcentration rate in all consumers is multiplied by a scaling factor that is between 0.2 and 2.0 with 0.2 intervals. The effect of this on the bioaccumulation in fish 2 for the Gotland Deep is shown to visualize the impact. Then the relative contribution of bioconcentration in consumers on the bioaccumulation of $MMHg^+$ in fish 2 is shown for all three setups. For the second sensitivity study, the same approach is used but the bioconcentration rate of producers is multiplied by a scaling factor.

**Author Response**

The results of this senstitivity study I would present in the results as follows:

**Suggested edit**

**Sensitivy of the consumer bioconcentration rate**

The results of the first sensitivity study, in which the bioconcentration rate of consumers is altered, are shown in Fig. 2. Figure 2a illustrates that the $MMHg^+$ contribution from bioconcentration in consumers is linearly related to the consumer bioconcentration rate scaling factor. Thus, altering the bioconcentration rate by half or double yields the same relative effect on fish 2's $MMHg^+$ content from direct bioconcentration. Based on Table 1, we can see that in the Gotland Deep, the difference between the simulation with and

without consumer bioconcentration is 0.0183 ng Hg mgC$^{-1}$. This means that picking a bioconcentration double the real rate would result in a 0.0183 ng Hg mgC$^{-1}$ overestimation of MMHg$^+$ bioaccumulation in fish 2, while selecting bioconcentration rates half the true values would result in a reduction of 0.00915 ng Hg mgC$^{-1}$. However, the relative contribution of bioconcentration to total MMHg$^+$ bioaccumulation follows a non-linear pattern, as shown in Fig. 2b. This non-linearity occurs because the total MMHg$^+$ in fish 2 is influenced by both bioconcentration in consumers and bioconcentration in producers. When the consumer bioconcentration scaling factor is 0, bioconcentration in consumers makes no contribution to fish 2's MMHg$^+$ levels. Conversely, this contribution can never reach 100% because bioconcentration in producers and consequent biomagnification from lower trophic levels always contributes to the total MMHg$^+$ burden in fish. In the same way as in the results shown in Table 1, the relative importance of bioconcentration is consequently highest in the Northern North Sea, followed by the Southern North Sea and lowest in the Gotland Deep.

**Sensitivy of the producer bioconcentration rate**

The results of the second sensitivity study are shown in Fig. 3. Here, rather than the consumer bioconcentration rate, the producers' bioconcentration rates are multiplied by a scaling factor. Again, the effect of this scaling on the bioaccumulation in all trophic levels is visualised in Fig. 3a, and the effect of this scaling on the relative importance of consumer bioconcentration on MMHg$^+$ bioaccumulation is shown in Fig. 3b. If the bioconcentration scaling factor is 0, there is still MMHg$^+$ bioaccumulation in fish 2, both from direct bioconcentration and from bioconcentration in consumers and consequent biomagnification. The increase in fish 2 MMHg$^+$ per step of 0.2 in the scaling factor is $0.0083 \pm 0.00030$ ng Hg mg$^{-1}$. The relative contribution of consumer bioconcentration on MMHg$^+$ bioaccumulation is shown in Fig. 3b. An important note here is that while we scaled the bioconcentration factor of producers and consumers, MMHg$^+$ can also be bioaccumulated via the partitioning to dissolved organic matter (DOM) detritus and consequent biomagnification. This is especially important in the Northern North Sea. In the seasonally stratified water column, macrobenthos cannot feed directly off the phytoplankton bloom; thus, the dying and sinking of particles is an important flux that is consumed by the benthos. Benthos, in turn, is an important food source for fish 2. So scaling the producer bioconcentration rate has less effect in the Northern North Sea. In the Gotland Deep, the opposite is true; because the deep water is anoxic, there is no macrobenthos in the model. This means that the entire ecosystem is pelagic and detritus is less important than direct consumption of the phytoplankton bloom.

[Figure]

Figure 2: a) show the effect of the bioaccumulation of MMHg$^+$ per trophic level in the Gotland Deep. This shows an increase $0.0036 \pm 0.00010$ ng Hg mg$^{-1}$ in fish 2 MMHg$^+$ bioaccumulation for every 0.2 step increase in the consumer bioconcentration scacling factor. 2b) shows the percentage difference due to bioaccumulation with different consumer bioconcentration scaling factors in all setups. GD refers to the Gotland Deep, NNS to the Northern North Sea and SNS to the Southern North Sea. When the consumers bioconcentration scaling factor is 0, the percentage difference due to bioconcentration is 0 %. As this increase the percentage increases. The relationship between the consumers bioconcentration factor and the percentage difference due to consumer bioconcentration is plotted assuming an saturating exponential relationship.

[Figure]

Figure 3: a) illustrates the influence of scaling the producers bioconcentration rate of MMHg$^+$ on the MMHg$^+$ bioaccumulation at each trophic level in the Gotland Deep. This shows an increase of $0.0084 \pm 0.00032$ ng Hg mg$^{-1}$ in fish 2 MMHg$^+$ with every 0.2 increase in the producers scaling factor. 3b) highlights the difference due to consumer bioconcentration with different primary producers scalings factors across all setups. The relationship between producer bioconcentration scaling factor and the percentage difference in MMHg$^+$ bioaccumulation in fish 2 due to consumer bioconcentration is plotted using an expenontial decay function. This shows that in all cases the percentage difference is high when the producer bioconcentration factor is 0, and that this percentages decreases with an increasing scaling factor.

> **Author Response**
>
> An additionaal component of the overal uncertainity is the uncertainty of the rest of the system, for example phytoplankton size distribution. Completley assesing this uncertainty is not really possible as our North and Baltic Seas setups cannot represent conditions that might be present in other areas. To adresss this I suggest I suggest I estimate the upper and lower bounds of the importance of the bioconcentration of MMHg$^+$ based on field observations and supplement the sensitivity analyses provided with this more general discussion arround the expected upper and lower bounds that are expected. To do this I would suggest to add the following part to the end of the Model Limitations section, on line 302.

**Suggested edit**

The results of our model represent just one possible outcome based on a regional setup representing the North and Baltic Seas, and the importance of bioconcentration can vary greatly depending on the bioconcentration factors of all species in the trophic chain. We can asses expected range of importance of consumer level bioconcentration by developing theoretical maximum and minimum values based on observational studies. We can estimate that direct bioconcentration in zooplankton may account for up to 50%, based on Lee and Fisher (2017), and similarly for mid-trophic level fish, based on Wang and Wong (2003).

We can use this to estimate the maximum expected contribution of consumer level bioconcntration on bioaccumulation by making two assumptions: (1) bioconcentration in both copepods and fish lies between 0 and 50% and is equal across all trophic levels, and (2) the food chain is linear, meaning that trophic level 3 feeds exclusively on trophic level 2, which feeds exclusively on trophic level 1. Under these assumptions, we can estimate the percentage of $MMHg^+$ in the diet of a given trophic level that originated from bioconcentration in primary producers as:

$$PBC\%_n = (1 - BC)^{n-1} \times 100\% \tag{8}$$

where:

- $PBC\%_n$ is the percentage of $MMHg^+$ in the diet of trophic level $n$ that originates from bioconcentration at the primary producer level,

- $BC$ is the fraction (0–1) of $MMHg^+$ at each trophic level originating from bioconcentration.

Although this is a simplification, it illustrates that a high bioconcentration estimate of 50% results in only 12.5% of $MMHg^+$ in the diet of a trophic level 4 fish originating from bioconcentration in primary producers, meaning that 87.5% originates from consumer-level processes. Even a low estimate of 10% still results in 27.1% of $MMHg^+$ in the diet of the same high-trophic-level fish originating from consumer-level bioconcentration.

The degree to which this interaction contributes to overall bioaccumulation depends on numerous additional factors that are not yet fully understood, including the size distribution of phytoplankton at the base of the food web, the trophic structure, consumer metabolic and respiration rates, and the assimilation efficiency of $MMHg^+$ from the diet. This complexity makes it difficult, if not impossible, to provide a definitive estimate of the importance of consumer-level bioconcentration and the uncertainity of the interaction. However, based on the bioconcentration rates provided in the current literature, we conclude that this process plays a key role the bioaccumulation of $MMHg^+$ in higher trophic levels.

---

## Author Comment (AC2)

**Author Response**

Thank your excellent feedback. Below, I will discuss how I would implement all of the suggested comments.

**Reviewer Comment**

- Section 2.2: Are there any equations for the parameterization schemes in the bioaccumulation of Hg in the model? Model equations are important for understanding the critical processes of the substance. Meanwhile, what are the critical parameters and coefficients for the critical processes in the model? The details of this model are not clarified in the method.

- Significantly, model performance should be evaluated against observations, which are deficient in this study. The literature Amptmeijer et al. (2025) is very important for this study. However, we cannot access to the paper because it is in preparation.

**Author Response**

My apologies. The publication of a key paper for this model was delayed. These details of this model are referenced in Amptmeijer et al. 2025, which is currently available with https://doi.org/10.5194/egusphere-2025-1486 I believe this is the main concern of the 3 points above. Here, the key equations and validation of both carbon fluxes and bioaccumulation are discussed in more detail. The model described and evaluated in Amptmeijer et al. 2025 is run with and without the bioaccumulation of Hg2+ and with and without the bioaccumulation of MMHg+ in consumers. I addition to this paper now beign available I would discuss in more detail exactly which equations we use.

**Suggested edit**

**Used terminology: bioaccumulation, bioconcentration, biomagnification, and *in vivo* Hg methylation**

**Bioaccumulation** in the marine environment refers to the total increase in pollutants in biota compared to that in the water. This can be quantified in nature by measuring the concentration of pollutants in both water and biota and estimating the difference. This is typically expressed as the bioaccumulation factor, BAF. For example, the bioaccumulation of MMHg$^+$ in organisms $i$ can be calculated based on observations as:

$$\text{BAF}_i^{MMHg^+} = \frac{C_i^{MMHg^+}}{C_w^{MMHg^+}} \tag{1}$$

In which,

$$\text{BAF}_i^{\text{MMHg}^+} = \text{The bioaccumulation factor of MMHg}^+ \text{ for organism } i \, [\text{L} \cdot \text{kg}^{-1}]$$
$$C_i^{\text{MMHg}^+} = \text{The concentration of MMHg}^+ \text{ in organism } i \, [\text{ng Hg} \cdot \text{kg}^{-1}]$$
$$C_w^{\text{MMHg}^+} = \text{The free concentration of MMHg}^+ \text{ in water } [\text{ng Hg} \cdot \text{L}^{-1}]$$

Since the BAF can be based on field measurements, it is a commonly used metric to estimate the link between the concentrations of pollutants in seawater and those in biota. In this study, we are interested in separating the bioaccumulation into separate pathways: the direct uptake from the water (bioconcentration) and the increase in pollutants due to trophic interactions (biomagnification).

**Bioconcentration**, is the increase in the concentration of Hg in biota directly due to uptake from the water. Because the process of bioconcentration relies on the exchange of Hg between the dissolved phase and an organism, it depends on the surface area of the organic material that is in contact with the water. Due to this, small organisms, such as bacteria and phytoplankton, have a greater ability to bioconcentrate Hg (Mason et al., 1996; Pickhardt et al., 2006). However, the bioconcentration process is complicated and recent studies show that the bioconcentration of $MMHg^+$ is influenced by cell-dependent factors, such as the thickness of the phycosphere and the availability of transmembrane channels, while this is not the case for $Hg^{2+}$ (Garcia-Arevalo et al., 2024). Bioconcentration is typically defined by the bioconcentration factor (BCF). The BCF for $MMHg^+$ in organism $i$, can for example be calculated as

$$\text{BCF}_i^{Hg} = \frac{BC_i^{Hg}}{C_w^{Hg}} \tag{2}$$

In which,

$$\text{BCF}_i^{\text{Hg}} = \text{The bioconcentration factor of Hg for organism } i \, [\text{L} \cdot \text{kg}^{-1}]$$
$$BC_i^{\text{Hg}} = \text{The concentration of Hg in organism } i \text{ due to direct bioconcentration} \, [\text{ng Hg} \cdot \text{kg}^{-1}]$$
$$C_w^{\text{Hg}} = \text{The free concentration of Hg in the water} \, [\text{ng Hg} \cdot \text{L}^{-1}]$$

Here, Hg could either refer to $Hg^{2+}$ or $MMHg^+$. Note that this defines the theoretical BCF. In nature it is typically impossible to directly measure the BCF, as it would be impossible to separate between $MMHg^+$ that is taken up directly from the water and $MMHg^+$ that is ingested via food. Bioconcentration is the most important step in bioaccumulation and phytoplankton can have a BCF of $MMHg^+$ between 2E4 L kg$^{-1}$ and 6.4E6 L kg$^{-1}$ (Gosnell & Mason, 2015).

**Biomagnification** is when $MMHg^+$ reaches higher concentrations at progressively higher trophic levels. The biomagnification factor, the fractional increase in $MMHg^+$ with each trophic level, is estimated to be $7.0 \pm 4.9$ (Harding et al., 2018; Lavoie et al., 2013). This means that in addition to the high increase in $MMHg^+$ in phytoplankton, there is a large increase in $MMHg^+$ at every consecutive trophic level. Many seafoods consist of high-trophic animals, such as cod, tuna, or marlin, which can have trophic levels between 4 and 4.8 (Nilsen et al., 2008; Sarà & Sarà, 2007). Biomagnification can increase the already high levels of $MMHg^+$ in phytoplankton by up to another factor $11.9^{4.8} \approx 145420$. This is typically defined by the biomagnification factor, BMF, which can be calculated assuming steady state for organism i, preying on organism j for $MMHg^+$ as:

$$\text{BMF}_{i,j}^{Hg} = \frac{C_i^{Hg}}{C_j^{Hg}} \tag{3}$$

In which,

$$\text{BMF}_{ij}^{\text{Hg}} = \text{The biomagnification factor for trophic consumption of organism } j \text{ by } i \, [\text{unitless}]$$
$$C_j^{\text{Hg}} = \text{The concentration of Hg in organism } j \, [\text{ng Hg} \cdot \text{kg}^{-1}]$$
$$C_i^{\text{Hg}} = \text{The concentration of Hg in organism } i \, [\text{ng Hg} \cdot \text{kg}^{-1}]$$

The biomagnification factor of $MMHg^+$ is extremely high, Lavoie et al. (2013) estimates the diet-weighted average BMF for $MMHg^+$ as $8.1 \pm 7.2$ while it is only $4.7 \pm 4.7$ for $Hg^{2+}$. This combined with the higher toxicity of $MMHg^+$ is the reason why the bioaccumulation of $MMHg^+$ is of much higher concern than the bioaccumulation of $Hg^{2+}$.

**In vivo methylation** occurs when animals take other forms of Hg and transform it into $MMHg^+$ in organisms. Although the existence of this process has been demonstrated in specific organisms such as cuttlefish, it is poorly understood and only recently gaining attention (Gente et al., 2023). There is no direct evidence of in vivo methylation in the animals that we model, so it is not implemented in this model.

Overall the dominant pathway of the bioaccumulation of $MMHg^+$ is the bioconcentration of $MMHg^+$ in phytoplankton and consequent biomagnification. The important route is quantified by Wu et al. (2019) using a meta-analysis. They find that the concentration of MeHg at the base of the food web predicts 63% of the observed variability in high trophic level fish, while the remaining 37% is controlled by factors such as the dissolved organic matter content and oligotrophy.

**Author Response**

I would suggest to expand section 2.2 in the methods section (line 184) to explain the exact equations used in this paper to asses bioconcentration and bioaccumulation.

**Suggested edit**

The implementation of bioaccumulation is discussed and validated in more detail in Amptmeijer et al. (2025), but the core equations are discussesed here as well for clarity. The increase in bioconcentrated pollutant ($Hg^{2+}$ or $MMHg^+$) per day for a functional group is calculated based on the biomass concentration of the group, the uptake rate, and the concentration of the pollutant, while it is reduced with a rate that is the sum of the release rate of the pollutant and the loss of biomass from group g, from both biologicall loss (respiration and mortality) and predation. The change in pollutant p due to bioaccumulation can then be calculated using the following equation:

$$\frac{dC_{g,p}^{BC}}{dt} = b_g \cdot C_p^{env} \cdot r_{g,p}^{bc} - C_{g,p}^{BC} \cdot (r_{g,p}^{rel} + r_g^{bl} + \sum_{z=1}^{n_z} r_{z,g}^{pred}) \qquad (4)$$

$$
\begin{aligned}
C_{g,p}^{BC} &= \text{Bioconcentrated pollutant } p \text{ in group } g \text{ [ng Hg m}^{-3}\text{]} \\
b_g &= \text{Biomass of functional group } g \text{ [mgC m}^{-3}\text{]} \\
C_p^{env} &= \text{Environmental concentration of pollutant } p \text{ [ng Hg m}^{-3}\text{]} \\
r_{g,p}^{bc} &= \text{Bioconcentration rate for group } g \text{ and pollutant } p \text{ [ng Hg mgC}^{-1}\text{ d}^{-1}\text{]} \\
r_{g,p}^{rel} &= \text{Release rate of pollutant } p \text{ from group } g \text{ [d}^{-1}\text{]} \\
r_g^{bl} &= \text{Biological loss rate for group } g \text{ (mortality, respiration) [d}^{-1}\text{]} \\
r_{z,g}^{pred} &= \text{Predation rate by predator } z \text{ on group } g \text{ [d}^{-1}\text{]} \\
n_z &= \text{Number of consumer groups feeding on group } g \\
z &= \text{Index for consumer groups (predators) of } g \\
t &= \text{Time [d]}
\end{aligned}
$$

While the change in pollutant p due to biomagnfication is also dependent on the predation and concentration of pollutants from both bioconcentration and biomagniciation in the prey. Additionally pollutant p is released via the turnover rate rather than the release rate as is the case for bioconcentration, the change in pollutant p due to biomagnification can then be calculated as follows:

$$\frac{dC_{g,p}^{BM}}{dt} = \sum_{s=1}^{n_s}(r_{g,s}^{pred} \cdot a_{s,p} \cdot (C_{s,p}^{BC} + C_{s,p}^{BM})) - C_{g,p}^{BM} \cdot (r_{p,g}^{to} + r_g^{bl}\sum_{z=1}^{n_z}r_{z,g}^{pred}) \qquad (5)$$

So the total concentration of pollutant P in ng Hg m$^{-3}$ is:

$$C_{(g,p)} = C_{(g,p)}^{BC} + C_{(g,p)}^{BM} \qquad (6)$$

Since this tracks the pollutants per volume of water, the total bioaccumulation per biomass in ng Hg mgC$^{-1}$ is then calculated as

$$C_{(g,p)}^{bg} = \frac{C_{(g,p)}}{b_g} \qquad (7)$$

This is then converted to the bioaccumulation per dry weight based on an assumed ratio of carbon to dry weight of 0.2 for diatoms, 0.33 for flagellates and cyanobacteria, and 0.5 for zooplankton and fish based on Walve and Larsson (1999) and Sicko-Goad et al. (1984).

**Author Response**

In addition, I would provide the key take conclusion of the model evaluation of this paper.I would place this after section 2.2 so at line 196 in the manuscript.

**Suggested edit**

**Performance of the GOTM-ECOSMO-MERCY model**

The model is generally consistent with observational data and the previously validated 3D ECOSMO E2E model in terms of biomass. Minor exception are that the Chlorophyll-a concentration in the Gotland Deep matches the Northern instead of the Central Baltic Sea, and that the fish biomass in the Gotland Deep is overestimated by 7% compared to Thurow (1997). The model also predicts tHg content and Hg$^{2+}$ and MMHg$^+$ levels in phytoplankton, zooplankton and fish 1 accurately, with MMHg+ bioaccumulation corresponding well with trophic interactions. A deviation is seen in the trophic level fish 2, which has a trophic level of 3.5–3.7 in the model, below the expected level for Atlantic Cod (4.0—4.2). Nonetheless, this level remains high making fish 2 representative of a high trophic level animals. The MMHg$^+$ bioaccumulation in fish 2 is consistent with the observed bioaccumulation for its trophic level. With the exceptions of the above discused minor exceptions, the model simulates biomass, Hg speciation and bioaccumulation in line with obsersations.

**Reviewer Comment**

The setup of the two scenarios is a sample. In my opinion, sensitivity analysis for critical parameters or uncertainty analysis of the results is needed. For the results and discussion, the illustrations are concise, and I cannot gain much in-depth discussion and thinking.

**Author Response**

I agree that the results of the paper are too limited, underexplored, and should be expanded upon in a sensitivity study. Additionally I would propse we plot the seasonality

of the interaction to create more depths in the analyses. I would propse to add to section 3.3 line 238 the following expansion of the results

**Suggested edit**

**Seasonality of the difference in MMHg$^+$ bioaccumulation**

The seasonality of the difference in MMHg$^+$ bioaccumulation caused the bioaccumulation of Hg$^{2+}$ and the bioconcentration of MMHg$^+$ in consumers is shown in Fig. 1. For each calendar day (January 1$^{st}$, January 2$^{nd}$, etc.), the modeled daily values from each of the last 10 years of the simulations were averaged. The resulting time series represents an annual cycle of average daily conditions. From the producers functional groups only the diatoms are shown as the reaction is not group specific but rather caused by changes in dissolved Hg$^{2+}$ and MMHg$^+$ which means the diffference caused for all phytoplankton groups was the same. This shows that, while the scale depends on the setup, there are interactions that consistently occur. In low trophic levels such as phytoplankotn and microzooplankton the bioaccumulation of Hg$^{2+}$ causes a seasonal respone in the MMHg$^+$ bioaccumulation in phytoplankton which is consequently observable in low trophic level biota such as microzooplankton. While this reduction in MMHg$^+$ would compound into higher trophic levels, its effects in higher trophic level animals dwarves in comparison to the difference caused by incorporating the bioconcentration of MMHg$^+$ in consumers and it does not cause a difference larger than 3% in either fish 1 or fish 2 in any of the setups.

[Figure]

Figure 1: The seasonality of the difference in the bioaccumulation of MMHg$^+$ caused by the bioaccumulation of Hg$^{2+}$ and the bioconcentration of MMHg in consumers for a) the Gotland Deep, b) the Northern North Sea and c) the Southern North Sea. In high trophic level such as fish 1 and fish 2 there is low seasonality and the effect of the bioconcentration of MMHg$^+$ in consumers is high while the effect of the bioaccumulation of Hg$^+$ is low. In low trophic levels, notably diatoms and microzooplankton there is strong seasonal component. The bioaccumulation of MMHg$^+$ is up to 5% lower in diatoms in the Southern North Sea if the bioaccumulation of Hg$^{2+}$ is modeled in late summer when biomass is high. But the bioaccumulation of Hg$^{2+}$ does not lead to a notable ($> 5\%$) difference at any moment in fish.

**Author Response**

I would propose that we do a sensitivity study for the first hypothesis, that bioconcentration in consumers is a significant driver of MeHg bioconcentration, by testing and visualizing the sensitivity of this process to both the bioconcentration rate of consumers

and of producers. I would run the model with 10 different bioconcentration factors between 0 (no biocentration) and 2.0 (double the estimated bioconcentration rate), for both consumers' and producers' bioconcentration. This would result in 20 different scenarios with varying bioconcentration rates to asses the sensitivity of importance of consumer level bioconcentration to the bioconcentration rates. I would suggest to add the following segment to form section 2.4 at line 209 in the methods as follows, in order to introduce the sensitivity study:

**Suggested edit**

In order to further investigate how bioconcentration in consumers affects bioaccumulation of $MMHg^+$, we performed a sensitivity analysis on the key drivers: the bioconcentration rate of consumers and the bioaccumulation rate of producers. To this extent, two sensitivity studies are performed. In the first sensitivity study, the bioconcentration rate in all consumers is multiplied by a scaling factor that is between 0.2 and 2.0 with 0.2 intervals. The effect of this on the bioaccumulation in fish 2 for the Gotland Deep is shown to visualize the impact. Then the relative contribution of bioconcentration in consumers on the bioaccumulation of $MMHg^+$ in fish 2 is shown for all three setups. For the second sensitivity study, the same approach is used but the bioconcentration rate of producers is multiplied by a scaling factor.

**Author Response**

And then present the results of this study as follow, by uncerting the part below at line 257 at section 3.6:

**Suggested edit**

**Sensitivy of the consumer bioconcentration rate**

The results of the first sensitivity study, in which the bioconcentration rate of consumers is altered, are shown in Fig. 2. Figure 2a illustrates that the $MMHg^+$ contribution from bioconcentration in consumers is linearly related to the consumer bioconcentration rate scaling factor. Thus, altering the bioconcentration rate by half or double yields the same relative effect on fish 2's $MMHg^+$ content from direct bioconcentration. Based on Table 1, we can see that in the Gotland Deep, the difference between the simulation with and without consumer bioconcentration is 0.0183 ng Hg mgC$^{-1}$. This means that picking a bioconcentration double the real rate would result in a 0.0183 ng Hg mgC$^{-1}$ overestimation of $MMHg^+$ bioaccumulation in fish 2, while selecting bioconcentration rates half the true values would result in a reduction of 0.00915 ng Hg mgC$^{-1}$. However, the relative contribution of bioconcentration to total $MMHg^+$ bioaccumulation follows a non-linear pattern, as shown in Fig. 2b. This non-linearity occurs because the total $MMHg^+$ in fish 2 is influenced by both bioconcentration in consumers and bioconcentration in producers. When the consumer bioconcentration scaling factor is 0, bioconcentration in consumers makes no contribution to fish 2's $MMHg^+$ levels. Conversely, this contribution can never reach 100% because bioconcentration in producers and consequent biomagnification from lower trophic levels always contributes to the total $MMHg^+$ burden in fish. In the same way as in the results shown in Table 1, the relative importance of bioconcentration is consequently highest in the Northern North Sea, followed by the Southern North Sea and lowest in the Gotland Deep.

**Sensitivy of the producer bioconcentration rate**

The results of the second sensitivity study are shown in Fig. 3. Here, rather than the consumer bioconcentration rate, the producers' bioconcentration rates are multiplied by a scaling factor. Again, the effect of this scaling on the bioaccumulation in all trophic levels is visualised in Fig. 3a, and the effect of this scaling on the relative importance of consumer bioconcentration on MMHg$^+$ bioaccumulation is shown in Fig. 3b. If the bioconcentration scaling factor is 0, there is still MMHg$^+$ bioaccumulation in fish 2, both from direct bioconcentration and from bioconcentration in consumers and consequent biomagnification. The increase in fish 2 MMHg$^+$ per step of 0.2 in the scaling factor is $0.0083 \pm 0.00030$ ng Hg mg$^{-1}$. The relative contribution of consumer bioconcentration on MMHg$^+$ bioaccumulation is shown in Fig. 3b. An important note here is that while we scaled the bioconcentration factor of producers and consumers, MMHg$^+$ can also be bioaccumulated via the partitioning to dissolved organic matter (DOM) detritus and consequent biomagnification. This is especially important in the Northern North Sea. In the seasonally stratified water column, macrobenthos cannot feed directly off the phytoplankton bloom; thus, the dying and sinking of particles is an important flux that is consumed by the benthos. Benthos, in turn, is an important food source for fish 2. So scaling the producer bioconcentration rate has less effect in the Northern North Sea. In the Gotland Deep, the opposite is true; because the deep water is anoxic, there is no macrobenthos in the model. This means that the entire ecosystem is pelagic and detritus is less important than direct consumption of the phytoplankton bloom.

[Figure]

Figure 2: a) show the effect of the bioaccumulation of MMHg$^+$ per trophic level in the Gotland Deep. This shows an increase $0.0036 \pm 0.00010$ ng Hg mg$^{-1}$ in fish 2 MMHg$^+$ bioaccumulation for every 0.2 step increase in the consumer bioconcentration scacling factor. 2b) shows the percentage difference due to bioaccumulation with different consumer bioconcentration scaling factors in all setups. GD refers to the Gotland Deep, NNS to the Northern North Sea and SNS to the Southern North Sea. When the consumers bioconcentration scaling factor is 0, the percentage difference due to bioconcentration is 0 %. As this increase the percentage increases. The relationship between the consumers bioconcentration factor and the percentage difference due to consumer bioconcentration is plotted assuming an saturating exponential relationship.

[Figure]

Figure 3: a) illustrates the influence of scaling the producers bioconcentration rate of MMHg$^+$ on the MMHg$^+$ bioaccumulation at each trophic level in the Gotland Deep. This shows an increase of $0.0084 \pm 0.00032$ ng Hg mg$^{-1}$ in fish 2 MMHg$^+$ with every 0.2 increase in the producers scaling factor. 3b) highlights the difference due to consumer bioconcentration with different primary producers scalings factors across all setups. The relationship between producer bioconcentration scaling factor and the percentage difference in MMHg$^+$ bioaccumulation in fish 2 due to consumer bioconcentration is plotted using an expenontial decay function. This shows that in all cases the percentage difference is high when the producer bioconcentration factor is 0, and that this percentages decreases with an increasing scaling factor.
* * *
**Author Response**

Evaluating the total uncertainty in the conclusion would be beneficial. However, a challenge is that fully assessing this uncertainty is not feasible because it depends on factors like the size distribution of phyto- and zooplankton and the food web structure, which vary regionally and temporally, and are beyond the scope of the North and Baltic Sea model used in this study. To adresss this I suggest I estimate the upper and lower bounds of the importance of the bioconcentration of MMHg$^+$ based on field observations. to do this I would suggest to add the following part to the end of the Model Limitations section, on line 302.
* * *
**Suggested edit**

The results of our model represent just one possible outcome based on a regional setup representing the North and Baltic Seas, and the importance of bioconcentration can vary greatly depending on the bioconcentration factors of all species in the trophic chain. We can asses expected range of importance of consumer level bioconcentration by developing theoretical maximum and minimum values based on observational studies. We can estimate that direct bioconcentration in zooplankton may account for up to 50%, based on Lee and Fisher (2017), and similarly for mid-trophic level fish, based on Wang and Wong (2003).

We can use this to estimate the maximum expected contribution of consumer level bioconcntration on bioaccumulation by making two assumptions: (1) bioconcentration in both copepods and fish lies between 0 and 50% and is equal across all trophic levels, and (2) the food chain is linear, meaning that trophic level 3 feeds exclusively on trophic level 2, which feeds exclusively on trophic level 1. Under these assumptions, we can estimate the percentage of MMHg$^+$ in the diet of a given trophic level that originated from bioconcentration in primary producers as:

$$\text{PBC\%}_n = (1 - \text{BC})^{n-1} \times 100\% \tag{8}$$

where:

- PBC%$_n$ is the percentage of MMHg$^+$ in the diet of trophic level $n$ that originates from bioconcentration at the primary producer level,

- BC is the fraction (0–1) of MMHg$^+$ at each trophic level originating from bioconcentration.

Although this is a simplification, it illustrates that a high bioconcentration estimate of 50% results in only 12.5% of MMHg$^+$ in the diet of a trophic level 4 fish originating from bioconcentration in primary producers, meaning that 87.5% originates from consumer-level processes. Even a low estimate of 10% still results in 27.1% of MMHg$^+$ in the diet of the same high-trophic-level fish originating from consumer-level bioconcentration.

The degree to which this interaction contributes to overall bioaccumulation depends on numerous additional factors that are not yet fully understood, including the size distribution of phytoplankton at the base of the food web, the trophic structure, consumer metabolic and respiration rates, and the assimilation efficiency of MMHg$^+$ from the diet. This complexity makes it difficult, if not impossible, to provide a definitive estimate of the importance of consumer-level bioconcentration and the uncertainity of the interaction. However, based on the bioconcentration rates provided in the current literature, we conclude that this process plays a key role the bioaccumulation of MMHg$^+$ in higher trophic levels.

**References**

Amptmeijer, D. J., Bieser, J., Mikheeva, E., Daewel, U., & Schrum, C. (2025). Bioaccumulation as a driver of high MeHg in coastal Seas. *EGUsphere [preprint]*.

Garcia-Arevalo, I., Berard, J.-B., Bieser, J., Le Faucheur, S., Hubert, C., Lacour, T., Thomas, B., Cossa, D., & Knoery, J. (2024). Mercury Accumulation Pathways in a Model Marine Microalgae: Sorption, Uptake, and Partition Kinetics.

Gente, S., Minet, A., Lopes, C., Tessier, E., Gassie, C., Guyoneaud, R., Swarzenski, P. W., Bustamante, P., Metian, M., Amouroux, D., & Lacoue-Labarthe, T. (2023). In Vivo Mercury (De)Methylation Metabolism in Cephalopods under Different pCO 2 Scenarios. *Cite This: Environ. Sci. Technol*, *57*, 5770.

Gosnell, K. J., & Mason, R. P. (2015). Mercury and methylmercury incidence and bioaccumulation in plankton from the central Pacific Ocean. *Marine Chemistry*, *177*, 772–780.

Harding, G., Dalziel, J., & Vass, P. (2018). Bioaccumulation of methylmercury within the marine food web of the outer Bay of Fundy, Gulf of Maine. *PLoS ONE*, *13*(7).

Lavoie, R. A., Jardine, T. D., Chumchal, M. M., Kidd, K. A., & Campbell, L. M. (2013). Biomagnification of Mercury in Aquatic Food Webs: A Worldwide Meta-Analysis.

Lee, C. S., & Fisher, N. S. (2017). Bioaccumulation of methylmercury in a marine copepod. *Environmental toxicology and chemistry*, *36*(5), 1287.

Mason, R. P., Reinfelder, J. R., & Morel, F. M. (1996). Uptake, toxicity, and trophic transfer of mercury in a coastal diatom. *Environmental Science and Technology*, *30*(6), 1835–1845.

Nilsen, M., Pedersen, T., Nilssen, E. M., & Fredriksen, S. (2008). Trophic studies in a high-latitude fjord ecosystem — a comparison of stable isotope analyses ($\delta$13C and $\delta$15N) and trophic-level estimates from a mass-balance model. *https://doi.org/10.1139/F08-180*, *65*(12), 2791–2806.

Pickhardt, P. C., Stepanova, M., & Fisher, N. S. (2006). Contrasting uptake routes and tissue distributions of inorganic and methylmercury in mosquitofish (Gambusia affinis) and redear sunfish (Lepomis microlophus). *Environmental Toxicology and Chemistry*, *25*(8), 2132–2142.

Sarà, G., & Sarà, R. (2007). Feeding habits and trophic levels of bluefin tuna Thunnus thynnus of different size classes in the Mediterranean Sea. *Journal of Applied Ichthyology*, *23*(2), 122–127.

Sicko-Goad, L. M., Schelske, C. L., & Stoermer, E. F. (1984). Estimation of intracellular carbon and silica content of diatoms from natural assemblages using morphometric techniques'. *Limnol. Oceanogr*, *29*(6), 1170–178.

Thurow, F. (1997). Estimation of the total fish biomass in the Baltic Sea during the 20th century. *ICES Journal of Marine Science*, *54*, 444–461.

Walve, J., & Larsson, U. (1999). Carbon, nitrogen and phosphorus stoichiometry of crustacean zooplankton in the Baltic Sea: implications for nutrient recycling.

Wang, W., & Wong, R. (2003). Bioaccumulation kinetics and exposure pathways of inorganic mercury and methylmercury in a marine fish, the sweetlips Plectorhinchus gibbosus. *Marine Ecology Progress Series*, *261*.

Wu, P., Kainz, M. J., Bravo, A. G., Åkerblom, S., Sonesten, L., & Bishop, K. (2019). The importance of bioconcentration into the pelagic food web base for methylmercury biomagnification: A meta-analysis. *Science of the Total Environment*, *646*, 357–367.

---

## Author Comment (AC3)

**Author Response**

Thank your excellent feedback. Below, I will discuss how I would implement all of the suggested comments.

**Reviewer Comment**

This study employed different models to identify the contributions of bioconcentration and biomagnification to Hg and MeHg bioaccumulation. The primary concern is that the description of the model applied and the data sources lack clarity.

**Author Response**

Before going into the specific comments, I would answer your general feedback. The model does not use field data and is solely based on a model. The core model cited is presented in Amptmeijer et al. (2025). Which is discussed and evaluated here in more detail: https://doi.org/10.5194/egusphere-2025-1486 The model presented by Amptmeijer et al. (2025) is then used to run with and without bioaccumulation of $Hg^{2+}$ and consumer-level bioconcentration of $MMHg^+$ to estimate the importance of these interactions. As such, the core message of the paper is aimed at showing the importance of these interactions on the outcome of the model, which shows that these interactions should be included in MMHg+ bioaccumulation models. I would expand the Methods section to include the exact bioaccumulation equations used as follows:

**Suggested edit**

**Used terminology: bioaccumulation, bioconcentration, biomagnification, and *in vivo* Hg methylation**

**Bioaccumulation** in the marine environment refers to the total increase in pollutants in biota compared to that in the water. This can be quantified in nature by measuring the concentration of pollutants in both water and biota and estimating the difference. This is typically expressed as the bioaccumulation factor, BAF. For example, the bioaccumulation of $MMHg^+$ in organisms $i$ can be calculated based on observations as:

$$\text{BAF}_i^{MMHg^+} = \frac{C_i^{MMHg^+}}{C_w^{MMHg^+}} \tag{1}$$

In which,

$$\text{BAF}_i^{\text{MMHg}^+} = \text{The bioaccumulation factor of MMHg}^+ \text{ for organism } i \,[\text{L} \cdot \text{kg}^{-1}]$$
$$C_i^{\text{MMHg}^+} = \text{The concentration of MMHg}^+ \text{ in organism } i \,[\text{ng Hg} \cdot \text{kg}^{-1}]$$
$$C_w^{\text{MMHg}^+} = \text{The free concentration of MMHg}^+ \text{ in water} \,[\text{ng Hg} \cdot \text{L}^{-1}]$$

Since the BAF can be based on field measurements, it is a commonly used metric to estimate the link between the concentrations of pollutants in seawater and those in biota. In this study, we are interested in separating the bioaccumulation into separate pathways: the direct uptake from the water (bioconcentration) and the increase in pollutants due to trophic interactions (biomagnification).

**Bioconcentration**, is the increase in the concentration of Hg in biota directly due to uptake from the water. Because the process of bioconcentration relies on the exchange of Hg between the dissolved phase and an organism, it depends on the surface area of the organic material that is in contact with the water. Due to this, small organisms, such as

bacteria and phytoplankton, have a greater ability to bioconcentrate Hg (Mason et al., 1996; Pickhardt et al., 2006). However, the bioconcentration process is complicated and recent studies show that the bioconcentration of $MMHg^+$ is influenced by cell-dependent factors, such as the thickness of the phycosphere and the availability of transmembrane channels, while this is not the case for $Hg^{2+}$ (Garcia-Arevalo et al., 2024). Bioconcentration is typically defined by the bioconcentration factor (BCF). The BCF for $MMHg^+$ in organism $i$, can for example be calculated as

$$\text{BCF}_i^{Hg} = \frac{BC_i^{Hg}}{C_w^{Hg}} \tag{2}$$

In which,

$$\text{BCF}_i^{\text{Hg}} = \text{The bioconcentration factor of Hg for organism } i \, [\text{L} \cdot \text{kg}^{-1}]$$
$$BC_i^{\text{Hg}} = \text{The concentration of Hg in organism } i \text{ due to direct bioconcentration} \, [\text{ng Hg} \cdot \text{kg}^{-1}]$$
$$C_w^{\text{Hg}} = \text{The free concentration of Hg in the water} \, [\text{ng Hg} \cdot \text{L}^{-1}]$$

Here, Hg could either refer to $Hg^{2+}$ or $MMHg^+$. Note that this defines the theoretical BCF. In nature it is typically impossible to directly measure the BCF, as it would be impossible to separate between $MMHg^+$ that is taken up directly from the water and $MMHg^+$ that is ingested via food. Bioconcentration is the most important step in bioaccumulation and phytoplankton can have a BCF of $MMHg^+$ between 2E4 L kg$^{-1}$ and 6.4E6 L kg$^{-1}$ (Gosnell & Mason, 2015).

**Biomagnification** is when $MMHg^+$ reaches higher concentrations at progressively higher trophic levels. The biomagnification factor, the fractional increase in $MMHg^+$ with each trophic level, is estimated to be $7.0 \pm 4.9$ (Harding et al., 2018; Lavoie et al., 2013). This means that in addition to the high increase in $MMHg^+$ in phytoplankton, there is a large increase in $MMHg^+$ at every consecutive trophic level. Many seafoods consist of high-trophic animals, such as cod, tuna, or marlin, which can have trophic levels between 4 and 4.8 (Nilsen et al., 2008; Sarà & Sarà, 2007). Biomagnification can increase the already high levels of $MMHg^+$ in phytoplankton by up to another factor $11.9^{4.8} \approx 145420$. This is typically defined by the biomagnification factor, BMF, which can be calculated assuming steady state for organism i, preying on organism j for $MMHg^+$ as:

$$\text{BMF}_{i,j}^{Hg} = \frac{C_i^{Hg}}{C_j^{Hg}} \tag{3}$$

In which,

$$\text{BMF}_{ij}^{\text{Hg}} = \text{The biomagnification factor for trophic consumption of organism } j \text{ by } i \, [\text{unitless}]$$
$$C_j^{\text{Hg}} = \text{The concentration of Hg in organism } j \, [\text{ng Hg} \cdot \text{kg}^{-1}]$$
$$C_i^{\text{Hg}} = \text{The concentration of Hg in organism } i \, [\text{ng Hg} \cdot \text{kg}^{-1}]$$

The biomagnification factor of $MMHg^+$ is extremely high, Lavoie et al. (2013) estimates the diet-weighted average BMF for $MMHg^+$ as $8.1 \pm 7.2$ while it is only $4.7 \pm 4.7$ for $Hg^{2+}$. This combined with the higher toxicity of $MMHg^+$ is the reason why the bioaccumulation of $MMHg^+$ is of much higher concern than the bioaccumulation of $Hg^{2+}$.

**In vivo methylation** occurs when animals take other forms of Hg and transform it into $MMHg^+$ in organisms. Although the existence of this process has been demonstrated in specific organisms such as cuttlefish, it is poorly understood and only recently gaining

attention (Gente et al., 2023). There is no direct evidence of in vivo methylation in the animals that we model, so it is not implemented in this model.

Overall the dominant pathway of the bioaccumulation of $MMHg^+$ is the bioconcentration of $MMHg^+$ in phytoplankton and consequent biomagnification. The important route is quantified by Wu et al. (2019) using a meta-analysis. They find that the concentration of MeHg at the base of the food web predicts 63% of the observed variability in high trophic level fish, while the remaining 37% is controlled by factors such as the dissolved organic matter content and oligotrophy.

**Author Response**

I would suggest to expand section 2.2 in the methods section (line 184) to explain the exact equations used in this paper to asses bioconcentration and bioaccumulation.

**Suggested edit**

The implementation of bioaccumulation is discussed and validated in more detail in Amptmeijer et al. (2025), but the core equations are discussesed here as well for clarity. The increase in bioconcentrated pollutant ($Hg^{2+}$ or $MMHg^+$) per day for a functional group is calculated based on the biomass concentration of the group, the uptake rate, and the concentration of the pollutant, while it is reduced with a rate that is the sum of the release rate of the pollutant and the loss of biomass from group g, from both biologicall loss (respiration and mortality) and predation. The change in pollutant p due to bioaccumulation can then be calculated using the following equation:

$$\frac{dC_{g,p}^{BC}}{dt} = b_g \cdot C_p^{env} \cdot r_{g,p}^{bc} - C_{g,p}^{BC} \cdot (r_{g,p}^{rel} + r_g^{bl} + \sum_{z=1}^{n_z} r_{z,g}^{pred}) \tag{4}$$

$$C_{g,p}^{BC} = \text{Bioconcentrated pollutant } p \text{ in group } g \text{ [ng Hg m}^{-3}\text{]}$$
$$b_g = \text{Biomass of functional group } g \text{ [mgC m}^{-3}\text{]}$$
$$C_p^{env} = \text{Environmental concentration of pollutant } p \text{ [ng Hg m}^{-3}\text{]}$$
$$r_{g,p}^{bc} = \text{Bioconcentration rate for group } g \text{ and pollutant } p \text{ [ng Hg mgC}^{-1} \text{ d}^{-1}\text{]}$$
$$r_{g,p}^{rel} = \text{Release rate of pollutant } p \text{ from group } g \text{ [d}^{-1}\text{]}$$
$$r_g^{bl} = \text{Biological loss rate for group } g \text{ (mortality, respiration) [d}^{-1}\text{]}$$
$$r_{z,g}^{pred} = \text{Predation rate by predator } z \text{ on group } g \text{ [d}^{-1}\text{]}$$
$$n_z = \text{Number of consumer groups feeding on group } g$$
$$z = \text{Index for consumer groups (predators) of } g$$
$$t = \text{Time [d]}$$

While the change in pollutant p due to biomagnfication is also dependent on the predation and concentration of pollutants from both bioconcentration and biomagniciation in the prey. Additionally pollutant p is released via the turnover rate rather than the release rate as is the case for bioconcentration, the change in pollutant p due to biomagnification can then be calculated as follows:

$$\frac{dC_{g,p}^{BM}}{dt} = \sum_{s=1}^{n_s} (r_{g,s}^{pred} \cdot a_{s,p} \cdot (C_{s,p}^{BC} + C_{s,p}^{BM})) - C_{g,p}^{BM} \cdot (r_{p,g}^{to} + r_g^{bl} \sum_{z=1}^{n_z} r_{z,g}^{pred}) \tag{5}$$

So the total concentration of pollutant P in ng Hg m$^{-3}$ is:

$$C_{(g,p)} = C_{(g,p)}^{BC} + C_{(g,p)}^{BM} \tag{6}$$

Since this tracks the pollutants per volume of water, the total bioaccumulation per biomass in ng Hg mgC$^{-1}$ is then calculated as

$$C_{(g,p)}^{bg} = \frac{C_{(g,p)}}{b_g} \tag{7}$$

This is then converted to the bioaccumulation per dry weight based on an assumed ratio of carbon to dry weight of 0.2 for diatoms, 0.33 for flagellates and cyanobacteria, and 0.5 for zooplankton and fish based on Walve and Larsson (1999) and Sicko-Goad et al. (1984).
* * *
**Author Response**

In addition, I would provide the key take conclusion of the model evaluation of this paper. I would place this after section 2.2 so at line 196 in the manuscript.
* * *
**Suggested edit**

**Performance of the GOTM-ECOSMO-MERCY model**

The model is generally consistent with observational data and the previously validated 3D ECOSMO E2E model in terms of biomass. Minor exception are that the Chlorophyll-a concentration in the Gotland Deep matches the Northern instead of the Central Baltic Sea, and that the fish biomass in the Gotland Deep is overestimated by 7% compared to Thurow (1997). The model also predicts tHg content and Hg$^{2+}$ and MMHg$^+$ levels in phytoplankton, zooplankton and fish 1 accurately, with MMHg+ bioaccumulation corresponding well with trophic interactions. A deviation is seen in the trophic level fish 2, which has a trophic level of 3.5–3.7 in the model, below the expected level for Atlantic Cod (4.0—4.2). Nonetheless, this level remains high making fish 2 representative of a high trophic level animals. The MMHg$^+$ bioaccumulation in fish 2 is consistent with the observed bioaccumulation for its trophic level. Witthout the above discused minor exceptions, the model simulates biomass, Hg speciation and bioaccumulation in line with obsersations.
* * *
**Reviewer Comment**

Lines 6-9: This sentence is too long and not clear to me
* * *
**Author Response**

I would suggest that I rewrite the sentence as follows:
* * *
**Suggested edit**

In this study, we use a fully coupled 1D water column Hg bioaccumulation model to quantify how total bioaccumulation of Hg$^{2+}$ and uptake of MMHg$^+$ from the water (bioconcentration) in consumers affects the bioaccumulation of MMHg$^+$ in high trophic level fish. The study is performed in three setups representing hydrodynamic conditions representative of the North and Baltic Seas.

**Reviewer Comment**

Line 110: Descriptions about the Modeled region in the Introduction section is weird. I suggest moving it to the MM section.

**Author Response**

I can move the modeled regions section to the beginning of the introduction. I would put it at the beginning at line 149.

**Reviewer Comment**

The model section is not clear to me. How to divide bioconcentration and biomagnification. Is there any data collected from in-lab measurements?

**Reviewer Comment**

Table 1: What is the source of the data provided in this table?

**Author Response**

I would address these comments together as they addres the same issue. This study is purely model-based. Of course, previously published data collected from in-lab measurements are used to estimate the bioconcentration rates and assimilation efficiency, which drive the processes in our model. This is discussed in detail in the referenced model paper Amptmeijer and Bieser, 2025. Which is available here in preprint: https://doi.org/10.5194/egusphere-2025-1486. I agree that we underexplained the difference between bioconcentration and biomagnification and how this is done in our model. I hope the suggested expansion of the metods section described above adresses this concern. Additionally I would suggest to add the below statement at the beginning of the results section at line 210 to make sure there is no ambiguity about the source of the data.

**Suggested edit**

The results are presented in Table 1. All results are derived from model simulations. To quantify the influence of consumer level bioconcentration and and bioaccumulation of $Hg^{2+}$ on $MMHg^+$ bioaccumulation, the model was run under scenarios with and without bioaccumulation of $Hg^{2+}$ and with and without consumer-level bioconcentration of $MMHg^+$

---

## Author Comment (AC4)

**1 Answers to revieuwer 2**

**Reviewer Comment**

The introduction mentions "MMHg+is a topic of serious concern because MMHg+is a dangerous neurotoxin that can bioaccumulate to levels. that are dangerous for human consumption in fish that are often consumed as seafood." Could this phrase be a little more concise?

**Author Response**

I would suggest that we rewrite the sentence as follows:

**Suggested edit**

The element mercury (Hg) is presently included in the World Health Organization's list of the 10 substances of greatest concern (WHO, 2020). This is due to the capability of Hg to be methylated into monomethyl mercury ($MMHg^+$), a potent neurotoxin generated by microbial methylation of inorganic Hg. $MMHg^+$ biomagnifies within aquatic food webs, accumulating in predatory fish to concentrations that can impair human neurological development upon consumption.

**Reviewer Comment**

The introduction mentions "Since DMHg is susceptible to photodegradation, we can assume that it plays an important role in the coastal water investigated in this study, until better observational studies confirm or correct this assumption." There are problems with logic

**Author Response**

My apologies, that is indeed incorrect. I would correct it to:

**Suggested edit**

Given the rapid photodegradation of DMHg in natural water, DMHg is assumed not to significantly bioaccumulate in biota in the coastal area investigated in this study (West et al., 2022).

**Reviewer Comment**

The introduction mentions "the bioconcentration process is complicated and recent studies show that the bioconcentration of MMHg+is influenced by cell-dependent factors, such as the thickness of the phytosphere, while this is not the case for $Hg^{2+}$." How does the thickness of the phytosphere affect the bioconcentration of MMHg+? Why is Hg2+ not affected by this?

**Author Response**

Thank you for this comment. There is a lot of uncertainty about these processes and they are somewhat out of the scope of this modeling paper. Therefore I will suggest to enhance the unclear sentence with the expanded sentence belows, in order to create more clarity about this interaction while not distracting from the focus of the paper

**Suggested edit**

However, the bioconcentration process is controlled by a variety of factors, and recent studies show that the bioconcentration of $Hg^{2+}$ is constant when normalized for cell density, while the uptake of $MMHg^+$ is affected by changes in cell density and biomass. This suggests that $MMHg^+$ uptake is influenced by cell-dependent factors, such as the thickness of the phycosphere and the availability of transmembrane channels, while this is not the case for $Hg^{2+}$ (Garcia-Arevalo et al., 2024).

**Reviewer Comment**

The references cited in the manuscript are somewhat outdated and may benefit from incorporating more recent studies to ensure the relevance and accuracy of the presented information.

**Author Response**

I agree that especially in the segment about other models, there are new advancements made that should indeed be incorporated, and the novelty of this model against these papers should be evaluated. I would suggest to rewrite this section (line 62) as follows:

**Suggested edit**

Multiple models have been developed to explain $MMHg^+$ bioaccumulation in marine ecosystems. Key examples include trophic transfer (Schartup et al., 2018), base-level accumulation (Zhang et al., 2020), planktonic bioaccumulation in the Mediterranean Sea (Rosati et al., 2022), MeHg dynamics on the Beaufort Shelf (Li et al., 2022), and speciation plus accumulation in the North and Baltic Seas (Bieser et al., 2023).

In all previous models, the bioconcentration of $MMHg^+$ is included because it is an essential driver. These models, however, do not include higher trophic level animals such as fish. It is concluded in Schartup et al. (2018) that the bioconcentration of $MMHg^+$ in zooplankton is not a major contributor and contributes less than 15% of total MeHg bioaccumulation. Consequently, in later models such as presented by Rosati et al. (2022) this interaction is not included because their model focuses on the base of the food web. The study performed by Li et al. (2022) includes the process of bioconcentration for invertebrates, but it is not included for vertebrates. This means that our model would be the first model to include bioconcentration at every trophic level.

The bioaccumulation of $Hg^{2+}$ is much less studied and not incorporated in any of the above-mentioned models. This is because $Hg^{2+}$ is much less toxic than $MMHg^+$ and therefore comparably understudied. While data is limited, this raises the speculative question if the link between the bioaccumulation $Hg^{2+}$ and $MMHg^+$ is not underestimated as $Hg^{2+}$ and $MMHg^+$ are in active equilibrium in the water.

The ECOSMO-MERCY coupled system, which is used by Bieser et al. (2023) and Amptmeijer et al. (2025) is the only coupled model that models the bioaccumulation of $Hg^{2+}$ and $MMHg^+$ at higher trophic levels such as fish, while incorporating bioconcentration at every trophic level.

**Author Response**

Additionally, I would suggest to add the following statements based on modern references into the introduction. I would suggest to add the following statement to line 21:

**Suggested edit**

For example, it is estimated that the consumption of MeHg contaminated seafood contributed to 61,800 premature deaths and caused economic damage of up to 2.87 trillion USD (Chen et al., 2025). This issue is expected to become even more significant as antrhopgenic Hg emissions are projected to increase in the near future (Maria Brocza et al., 2024).

**Author Response**

And the below statement to line 45 in the manuscript:

**Suggested edit**

Additionally, it has been shown by Tesán-Onrubia et al. (2023) that plankton communities in the southern Mediteranean Sea have lower $MMHg^+$ concentration than plankton in the northern Mediterranean Sea, they linked this to changes in environmental conditions affecting bioconcentration.

**Author Response**

I also wanted to expand the theorical uncertainty of this interaction. Here I would also introduce some more up to date references. I would suggest to add the belows part to the model limitation section at line 302.

**Suggested edit**

The results of our model represent just one possible outcome based on a regional setup representing the North and Baltic Seas, and the importance of bioconcentration can vary greatly depending on the bioconcentration factors of all species in the trophic chain. We can asses expected range of importance of consumer level bioconcentration by developing theoretical maximum and minimum values based on observational studies. We can estimate that direct bioconcentration in zooplankton may account for up to 50%, based on Lee and Fisher (2017), and similarly for mid-trophic level fish, based on Wang and Wong (2003).

We can use this to estimate the maximum expected contribution of consumer level bioconcentration on bioaccumulation by making two assumptions: (1) bioconcentration in both copepods and fish lies between 0 and 50% and is equal across all trophic levels, and (2) the food chain is linear, meaning that trophic level 3 feeds exclusively on trophic level 2, which feeds exclusively on trophic level 1. Under these assumptions, we can estimate the percentage of $MMHg^+$ in the diet of a given trophic level that originated from bioconcentration in primary producers as:

$$\mathrm{PBC\%}_n = (1 - \mathrm{BC})^{n-1} \times 100\% \tag{1}$$

where:

- $\mathrm{PBC\%}_n$ is the percentage of $MMHg^+$ in the diet of trophic level $n$ that originates from bioconcentration at the primary producer level,

- BC is the fraction (0–1) of $MMHg^+$ at each trophic level originating from bioconcentration.

This is ofcourse a simplification. Huo et al. (2025) for example demonstrated that the

BMF varies with prey type, indicating that the % $MMHg^+$ from bioconcentration is not uniform across trophic levels. Furthermore, McClelland et al. (2024) observed higher $MMHg^+$ levels in animals feeding on benthic rather than pelagic invertebrates which is further supported by Liu et al. (2024) which found significantly higher $MMHg^+$ concentration in demersal fish compared to pelagic fish. Even if the BCF remains constant across different species, a higher BAF due to increased biomagnification would lessen the relative impact of bioconcentration. So the assumptions made in the above presented basic model are not true in nature, but despite this it can help us understand the strong importance of consumer level bioconcentration. This simplified model demonstrates that a 50% bioconcentration rate in trophic level 1, 2, and 3 leads to only 12.5% of $MMHg^+$ in the diet of a trophic level 4 fish being derived from primary producers, with the remaining 87.5% originating from consumer-level bioconcentration. Even a low estimate of 10% bioconcentration in trophic level 1,2, and 3 still results in 27.1% of $MMHg^+$ in the diet of trophic-level 4 fish originating from consumer-level bioconcentration.

The degree to which this interaction contributes to overall bioaccumulation depends on numerous additional factors that are not yet fully understood, including the size distribution of phytoplankton at the base of the food web, the trophic structure, consumer metabolic and respiration rates, and the assimilation efficiency of $MMHg^+$ from the diet. This complexity makes it difficult, if not impossible, to provide a definitive estimate of the importance of consumer-level bioconcentration and the uncertainty of the interaction. However, based on the bioconcentration rates provided in the current literature, we conclude that this process plays a key role in the bioaccumulation of $MMHg^+$ in higher trophic levels.

**Reviewer Comment**

The paper is very detailed about the basic knowledge and the content of the preliminary research, but the description of the later model establishment and the data obtained from the model is relatively brief, which can be further supplemented. For example, "The contribution of bioconcentration in zooplankton of 3.97-10.07%...... and the contribution of bioconcentration in fish between 8.14-21.82%......"

**Author Response**

Thank you for you comment. Based on suggestions of other researchers I have also expanded the interpretation of the results by analyzing the seasonality and performing sensitivity analyses. I would suggest to rewrite to start of the result (section 3 line 211) where I discuss the raw result. As below:

**Suggested edit**

**The biocentration of $MMHg^+$**

The direct contribution of bioconcentration varies greatly by functional group groups and setups. In zooplankton, the bioconcentration is minimally significant (4–10%), which is consistent with the model findings by Schartup et al. (2018) and the laboratory results by Lee and Fisher (2017), indicating that the bioconcentration in copepods represents up to 10% of total $MMHg^+$ bioaccumulation. In our model, the bioconcentration is highest in middle-level consumers: fish 1 (13–16%) and macrobenthos (14–25%), due to lower biological loss rates and relatively low dietary $MMHg^+$ concentrations. This is consistent with the study of (Wang & Wong, 2003) showing that bioconcentration accounts for 10–

50% of MMHg$^+$ in Sweetlips. In the modeled high-trophic level, fish 2, bioconcentration is a smaller relative component of total bioaccumulation, at 8—- 14%. Interestingly, mid-trophic level species show the highest relative bioconcentration contributions. The lower relative contribution in low-trophic-level animals is driven by a lower life expectancy and a higher organic turnover rate, which enhances the relative importance of dietary uptake. At the same time, the elevated MMHg$^+$ concentration in mid-trophic animals boosts dietary uptake in high-trophic level animals, compensating for reduced organic turnover in the top predators. On average of the three setups, the MMHg$^+$ uptake due to bioconcentratoin is 0.0053 ng Hg mg$^{-1}$ for fish 1 and 0.0055 ng Hg mg$^{-1}$ for fish 2. This means that fish 2 has a higher bioconcentration rate of MMHg$^+$ in absolute terms, but because its diet is richer in MMHg$^+$ compared to fish 1, the relative importance is lower.

**Author Response**

I would suggest to further enhance the result by discussion and visualing the seasonal component of the results as follows:

**Suggested edit**

**Seasonality of the difference in MMHg$^+$ bioaccumulation**

The seasonality of the difference in MMHg$^+$ bioaccumulation caused the bioaccumulation of Hg$^{2+}$ and the bioconcentration of MMHg$^+$ in consumers is shown in Fig. 1. To reduce the influence of interannual variability, we calculated a multi-year average of daily mean values over the last 10 years of the simulation. For each calendar day (January 1$^{st}$, January 2$^{nd}$, etc.), the modeled daily values from each of the last 10 years of the simulations were averaged. The resulting time series represents an annual cycle of average daily conditions. From the producers functional groups only the diatoms are shown as the reaction is not group specific but rather caused by changes in dissolved Hg$^{2+}$ and MMHg$^+$ which means the diffference caused for all phytoplankton groups was the same. This shows that, while the scale depends on the setup, there are interactions that consistently occur. In low trophic levels such as phytoplankotn and microzooplankton the bioaccumulation of Hg$^{2+}$ causes a seasonal respone in the MMHg$^+$ bioaccumulation in phytoplankton which is consequently observable in low trophic level biota such as microzooplankton. While this reduction in MMHg$^+$ would compound into higher trophic levels, its effects in higher trophic level animals dwarves in comparison to the difference caused by incorporating the bioconcentration of MMHg$^+$ in consumers and it does not cause a difference larger than 3% in either fish 1 or fish 2 in any of the setups.

[Figure]

Figure 1: The seasonality of the difference in the bioaccumulation of MMHg$^+$ caused by the bioaccumulation of Hg$^{2+}$ and the bioconcentration of MMHg in consumers for a) the Gotland Deep, b) the Northern North Sea and c) the Southern North Sea. In high trophic level such as fish 1 and fish 2 there is low seasonality and the effect of the bioconcentration of MMHg$^+$ in consumers is high while the effect of the bioaccumulation of Hg$^+$ is low. In low trophic levels, notably diatoms and microzooplankton there is strong seasonal component. The bioaccumulation of MMHg$^+$ is up to 5% lower in diatoms in the Southern North Sea if the bioaccumulation of Hg$^{2+}$ is modeled in late summer when biomass is high. But the bioaccumulation of Hg$^{2+}$ does not lead to a notable ($> 5\%$) difference at any moment in fish.

**Author Response**

Then I would expand the method section to introduce the senstivity study to expand the results as follows:

**Suggested edit**

In order to further investigate how bioconcentration in consumers affects bioaccumulation of MMHg$^+$, we performed a sensitivity analysis on the key drivers: the bioconcentration rate of consumers and the bioaccumulation rate of producers. To this extent, two sensitivity studies are performed. In the first sensitivity study, the bioconcentration rate in all consumers is multiplied by a scaling factor that is between 0.2 and 2.0 with 0.2 intervals. The effect of this on the bioaccumulation in fish 2 for the Gotland Deep is shown to visualize the impact. Then the relative contribution of bioconcentration in consumers on the bioaccumulation of MMHg$^+$ in fish 2 is shown for all three setups. For the second sensitivity study, the same approach is used but the bioconcentration rate of producers is multiplied by a scaling factor.

**Author Response**

The results of this senstitivity study I would present in the results as follows:

**Suggested edit**

**Sensitivy of the consumer bioconcentration rate**

The results of the first sensitivity study, in which the bioconcentration rate of consumers is altered, are shown in Fig. 2. Figure 2a illustrates that the MMHg$^+$ contribution from bioconcentration in consumers is linearly related to the consumer bioconcentration rate scaling factor. Thus, altering the bioconcentration rate by half or double yields the same

relative effect on fish 2's MMHg$^+$ content from direct bioconcentration. Based on Table 1, we can see that in the Gotland Deep, the difference between the simulation with and without consumer bioconcentration is 0.0183 ng Hg mgC$^{-1}$. This means that picking a bioconcentration double the real rate would result in a 0.0183 ng Hg mgC$^{-1}$ overestimation of MMHg$^+$ bioaccumulation in fish 2, while selecting bioconcentration rates half the true values would result in a reduction of 0.00915 ng Hg mgC$^{-1}$. However, the relative contribution of bioconcentration to total MMHg$^+$ bioaccumulation follows a non-linear pattern, as shown in Fig. 2b. This non-linearity occurs because the total MMHg$^+$ in fish 2 is influenced by both bioconcentration in consumers and bioconcentration in producers. When the consumer bioconcentration scaling factor is 0, bioconcentration in consumers makes no contribution to fish 2's MMHg$^+$ levels. Conversely, this contribution can never reach 100% because bioconcentration in producers and consequent biomagnification from lower trophic levels always contributes to the total MMHg$^+$ burden in fish. In the same way as in the results shown in Table 1, the relative importance of bioconcentration is consequently highest in the Northern North Sea, followed by the Southern North Sea and lowest in the Gotland Deep.

**Sensitivy of the producer bioconcentration rate**

The results of the second sensitivity study are shown in Fig. 3. Here, rather than the consumer bioconcentration rate, the producers' bioconcentration rates are multiplied by a scaling factor. Again, the effect of this scaling on the bioaccumulation in all trophic levels is visualised in Fig. 3a, and the effect of this scaling on the relative importance of consumer bioconcentration on MMHg$^+$ bioaccumulation is shown in Fig. 3b. If the bioconcentration scaling factor is 0, there is still MMHg$^+$ bioaccumulation in fish 2, both from direct bioconcentration and from bioconcentration in consumers and consequent biomagnification. The increase in fish 2 MMHg$^+$ per step of 0.2 in the scaling factor is $0.0083 \pm 0.00030$ ng Hg mg$^{-1}$. The relative contribution of consumer bioconcentration on MMHg$^+$ bioaccumulation is shown in Fig. 3b. An important note here is that while we scaled the bioconcentration factor of producers and consumers, MMHg$^+$ can also be bioaccumulated via the partitioning to dissolved organic matter (DOM) detritus and consequent biomagnification. This is especially important in the Northern North Sea. In the seasonally stratified water column, macrobenthos cannot feed directly off the phytoplankton bloom; thus, the dying and sinking of particles is an important flux that is consumed by the benthos. Benthos, in turn, is an important food source for fish 2. So scaling the producer bioconcentration rate has less effect in the Northern North Sea. In the Gotland Deep, the opposite is true; because the deep water is anoxic, there is no macrobenthos in the model. This means that the entire ecosystem is pelagic and detritus is less important than direct consumption of the phytoplankton bloom.

[Figure]

Figure 2: a) show the effect of the bioaccumulation of MMHg$^+$ per trophic level in the Gotland Deep. This shows an increase $0.0036 \pm 0.00010$ ng Hg mg$^{-1}$ in fish 2 MMHg$^+$ bioaccumulation for every 0.2 step increase in the consumer bioconcentration scacling factor. 2b) shows the percentage difference due to bioaccumulation with different consumer bioconcentration scaling factors in all setups. GD refers to the Gotland Deep, NNS to the Northern North Sea and SNS to the Southern North Sea. When the consumers bioconcentration scaling factor is 0, the percentage difference due to bioconcentration is 0 %. As this increase the percentage increases. The relationship between the consumers bioconcentration factor and the percentage difference due to consumer bioconcentration is plotted assuming an saturating exponential relationship.

[Figure]

Figure 3: a) illustrates the influence of scaling the producers bioconcentration rate of MMHg$^+$ on the MMHg$^+$ bioaccumulation at each trophic level in the Gotland Deep. This shows an increase of $0.0084 \pm 0.00032$ ng Hg mg$^{-1}$ in fish 2 MMHg$^+$ with every 0.2 increase in the producers scaling factor. 3b) highlights the difference due to consumer bioconcentration with different primary producers scalings factors across all setups. The relationship between producer bioconcentration scaling factor and the percentage difference in MMHg$^+$ bioaccumulation in fish 2 due to consumer bioconcentration is plotted using an expenontial decay function. This shows that in all cases the percentage difference is high when the producer bioconcentration factor is 0, and that this percentages decreases with an increasing scaling factor.

**References**

Amptmeijer, D. J., Bieser, J., Mikheeva, E., Daewel, U., & Schrum, C. (2025). Bioaccumulation as a driver of high MeHg in coastal Seas. *EGUsphere [preprint]*.

Bieser, J., Amptmeijer, D., Daewel, U., Kuss, J., Soerenson, A. L., & Schrum, C. (2023). The 3D biogeochemical marine mercury cycling model MERCY v2.0; linking atmospheric Hg to methyl mercury in fish. *Geoscientific Model Development Discussions*, 1–59.

Chen, Q., Wu, Q., Cui, Y., & Wang, S. (2025). Global seafood production practices and trade patterns contribute to disparities in exposure to methylmercury. *Nature Food 2025 6:5*, *6*(5), 491–502.

Garcia-Arevalo, I., Berard, J.-B., Bieser, J., Le Faucheur, S., Hubert, C., Lacour, T., Thomas, B., Cossa, D., & Knoery, J. (2024). Mercury Accumulation Pathways in a Model Marine Microalgae: Sorption, Uptake, and Partition Kinetics.

Huo, M., Pang, M., Ma, X., Wang, P., Sun, C., Zhang, Y., Gong, Y., Sun, Z., Zhang, Z., Wang, Z., Qu, P., Luo, X., Dutton, J., Bat, L., & Lei, P. (2025). Trophic transfer of mercury in marine food chains from the offshore waters of Changshan Archipelago OPEN ACCESS EDITED BY.

Lee, C. S., & Fisher, N. S. (2017). Bioaccumulation of methylmercury in a marine copepod. *Environmental toxicology and chemistry, 36*(5), 1287.

Li, M. L., Gillies, E. J., Briner, R., Hoover, C. A., Sora, K. J., Loseto, L. L., Walters, W. J., Cheung, W. W., & Giang, A. (2022). Investigating the dynamics of methylmercury bioaccumulation in the Beaufort Sea shelf food web: a modeling perspective. *Environmental Science: Processes & Impacts, 24*(7), 1010–1025.

Liu, T., An, M., Chen, J., Liu, Y., Chao, L., Liu, J., Zhang, M., Ogunseitan, O., Pestana, I., Peter Mason, R., & Soto-Jimenez, M. F. (2024). Variations in methylmercurycontamination levels andassociated health risks in differentfish species across three coastalbays in China. *Front. Environ. Sci, 12*, 1376882.

Maria Brocza, F., Rafaj, P., Sander, R., Wagner, F., & Marie Jones, J. (2024). Global scenarios of anthropogenic mercury emissions. *Atmos. Chem. Phys, 24*, 7385–7404.

McClelland, C., Chételat, J., Conlan, K., Aitken, A., Forbes, M. R., & Majewski, A. (2024). Methylmercury dietary pathways and bioaccumulation in Arctic benthic invertebrates of the Beaufort Sea. *Arctic Science, 10*(2), 305–320.

Rosati, G., Canu, D., Lazzari, P., & Solidoro, C. (2022). Assessing the spatial and temporal variability of methylmercury biogeochemistry and bioaccumulation in the Mediterranean Sea with a coupled 3D model. *Biogeosciences, 19*, 3663–3682.

Schartup, A. T., Qureshi, A., Dassuncao, C., Thackray, C. P., Harding, G., Sunderland, E. M., Harvard, †., & Paulson, J. A. (2018). A Model for Methylmercury Uptake and Trophic Transfer by Marine Plankton. *Environ. Sci. Technol, 52*, 18.

Tesán-Onrubia, J. A., Heimbürger-Boavida, L. E., Dufour, A., Harmelin-Vivien, M., García-Arévalo, I., Knoery, J., Thomas, B., Carlotti, F., Tedetti, M., & Bănaru, D. (2023). Bioconcentration, bioaccumulation and biomagnification of mercury in plankton of the Mediterranean Sea. *Marine Pollution Bulletin, 194*, 115439.

Wang, W., & Wong, R. (2003). Bioaccumulation kinetics and exposure pathways of inorganic mercury and methylmercury in a marine fish, the sweetlips Plectorhinchus gibbosus. *Marine Ecology Progress Series, 261*.

West, J., Gindorf, S., & Jonsson, S. (2022). Photochemical Degradation of Dimethylmercury in Natural Waters. *Environmental Science and Technology, 56*(9), 5920–5928.

Zhang, Y., Soerensen, A. L., Schartup, A. T., Sunderland, E. M., & Paulson, H. J. A. (2020). A Global Model for Methylmercury Formation and Uptake at the Base of Marine Food Webs. *Biogeochemical Cycles, 34*(2).

---

## Author Response (AR1)

**Introduction**

Below I will describe for each of the reviewers' comments how they were implemented. However, several reviewers asked for similar changes, of which some resulted in the rewriting of sections of the manuscript. I will post the rewritten manuscript sections here above and then reference that in the specific reference answers to not have to overinflate the document by posting the same rewritten sections several times.

**Implementation**

**Line 41**

Used terminology: bioaccumulation, bioconcentration, biomagnification, and *in vivo* Hg speciation

**Bioaccumulation** in the marine environment refers to the total increase in pollutants in biota compared to that in the water. This can be quantified in nature by measuring the concentration of pollutants in both water and biota and calculating the ratio. This is typically expressed as the bioaccumulation factor, BAF. For example, the bioaccumulation of  $MMHg^+$  in organisms i can be calculated based on observations as:

$$BAF_i^{MMHg^+} = \frac{C_i^{MMHg^+}}{C_w^{MMHg^+}} \tag{1}$$

In which,

$${\rm BAF}_i^{\rm MMHg^+} = {\rm The~bioaccumulation~factor~of~MMHg^+~for~organism~} i \, [{\rm L} \cdot {\rm kg}^{-1}] \qquad (2)$$

$$C_i^{\text{MMHg}^+} = \text{The concentration of MMHg}^+ \text{ in organism } i [\text{ng Hg} \cdot \text{kg}^{-1}]$$
 (3)

$$C_w^{\text{MMHg}^+}$$
 = The free concentration of MMHg+ in water [ng Hg · L-1] (4)

Since the BAF can be based on field measurements, it is a commonly used metric to estimate the link between the concentrations of pollutants in seawater and those in biota. In this study, we are interested in separating the bioaccumulation into separate pathways: the direct uptake from the water (bioconcentration) and the increase in pollutants due to trophic interactions (biomagnification).

Bioconcentration, is the increase in the concentration of Hg in biota directly due to uptake from the water. Because the process of bioconcentration relies on the exchange of Hg between the dissolved phase and an organism, it depends on the surface area of the organic material that is in contact with the water. Because of this, small organisms, such as bacteria and phytoplankton, have a greater ability to bioconcentrate Hg (Mason et al., 1996; Pickhardt et al., 2006). However, the bioconcentration process is controlled by a variety of factors, and recent studies show that the bioconcentration of  $Hg^{2+}$  is constant when normalized for cell density, while the uptake of MMHg+ is affected by changes in cell density and biomass. This suggests that MMHg+ uptake is influenced by cell-dependent factors, such as the thickness of the phycosphere and the availability of transmembrane channels, while this is not the case for  $Hg^{2+}$  (Garcia-Arevalo et al., 2024). Bioconcentration is typically defined by the bioconcentration factor (BCF). The BCF for MMHg+ in organisms i can be calculated as

$$BCF_i^{MMHg^+} = \frac{BC_i^{MMHg^+}}{C_w^{MMHg^+}}$$
 (5)

In which,

$$\mathrm{BCF}_{i}^{\mathrm{MMHg}^{+}} = \mathrm{The\ bioconcentration\ factor\ of\ MMHg}^{+}\ \text{for\ organism}\ i\left[\mathrm{L\cdot kg}^{-1}\right]$$
 (6)

 $BC_i^{\text{MMHg}^+}$  = The concentration of MMHg+ in organism *i* due to direct uptake from water [ng Hg·kg-1] (7)

$$C_w^{\text{MMHg}^+}$$
 = The free concentration of MMHg+ in the water [ng Hg · L-1] (8)

Here, Hg could either refer to  $\mathrm{Hg^{2+}}$  or  $\mathrm{MMHg^{+}}$ . Note that for consumers this would define the theoretical BCF. In nature it is typically only possible to measure the BCF in primary producers, as in consumers it would be impossible to separate between  $\mathrm{MMHg^{+}}$  that is taken up directly from the water and  $\mathrm{MMHg^{+}}$  that is ingested via food. Bioconcentration is the most important step in bioaccumulation and phytoplankton can have a BCF of  $\mathrm{MMHg^{+}}$  between  $2 \cdot 10^4$  L kg-1 and  $6.4 \cdot 10^6$  L kg-1 (Gosnell & Mason, 2015).

Biomagnification is when MMHg+ reaches higher concentrations at progressively higher trophic levels. The biomagnification factor, the fractional increase in MMHg+ with each trophic level, is estimated to be  $7.0 \pm 4.9$  (Harding et al., 2018; Lavoie et al., 2013). This means that in addition to the high concentration in MMHg+ in phytoplankton, there is a large increase in MMHg+ at every consecutive trophic level. Many seafoods consist of high-trophic animals, such as cod, tuna, or marlin, which can have trophic levels between 4 and 4.8 (Nilsen et al., 2008; Sarà & Sarà, 2007). Biomagnification can increase the already high levels of MMHg+ in phytoplankton by up to another factor  $11.9^{4.8} \approx 145420$ . This is typically defined by the biomagnification factor, BMF, which can be calculated assuming steady state for organism i, preying on organism j for MMHg+ as:

$$BMF_{i,j}^{MMHg^+} = \frac{C_i^{MMHg^+}}{C_j^{MMHg^+}}$$

$$\tag{9}$$

In which,

 $\mathrm{BMF}_{ij}^{\mathrm{MMHg}^{+}} = \mathrm{The~biomagnification~factor~for~trophic~consumption~of~organism~} j~\mathrm{by}~i\,[\mathrm{unitless}]~~(10)$

$$C_j^{\text{MMHg}^+} = \text{The concentration of MMHg}^+ \text{ in organism } j [\text{ng Hg} \cdot \text{kg}^{-1}]$$
 (11)

$$C_i^{\text{MMHg}^+} = \text{The concentration of MMHg}^+ \text{ in organism } i [\text{ng Hg} \cdot \text{kg}^{-1}]$$
 (12)

The biomagnification factor of MMHg+ is extremely high, Lavoie et al. (2013) estimates the diet-weighted average BMF in marine samples for MMHg+ as  $7.0 \pm 4.9$  while it is below 1 for iHg in most cases (Lavoie et al., 2013; Seixas et al., 2014). This combined with the higher toxicity of MMHg+ is the reason why the bioaccumulation of MMHg+ is of much higher concern than the bioaccumulation of Hg2+.

In vivo Hg speciation occurs when Hg is transformed from one form of Hg, such as  $MMHg^+$  into another form of Hg, such as  $Hg^{2+}$  in organisms. Although the existence of this process has been demonstrated in specific organisms such as cuttlefish, it is poorly understood and only recently gained attention (Gente et al., 2023). There is no direct evidence of *in vivo* methylation in the animals that we model, so it is not implemented in this model. But the relevance of *in vivo* Hg speciation cannot be excluded.

Overall the dominant pathway of bioaccumulation, the bioaccumulation of MMHg+ is the bioconcentration of MMHg+ in phytoplankton and consequent biomagnification. The importance of this route is quantified by Wu et al. (2019) using a meta-analysis. They find that the concentration of MeHg at the base of the food web predicts 63% of the observed variability in high-trophic-level fish, while the remaining 37% is controlled by factors such as the Dissolved Organic Matter (DOM) content and oligotrophy.

**Implementation**

**Line 108**

**Current models**

Multiple models have been developed to explain MMHg+ bioaccumulation in marine ecosystems. Key examples include trophic transfer (Schartup et al., 2018), base-level accumulation (Zhang et al., 2020), planktonic bioaccumulation in the Mediterranean Sea (Rosati et al., 2022), MeHg dynamics on the Beaufort Shelf (Li et al., 2022), and speciation and bioaccumulation in the North and Baltic Seas (Bieser et al., 2023).

In all of the previous models, bioconcentration of MMHg+ is included as it is an essential driver. It is concluded in Schartup et al. (2018) that the bioconcentration of MMHg+ in zooplankton contributes less than 15% of total MeHg bioaccumulation. Consequently, in later models such as presented by Rosati et al. (2022) this interaction is not included because their model focuses on the base of the food web. The study performed by Li et al. (2022) includes the process of bioconcentration for invertebrates, but it is not included for vertebrates. This means that our model would be the first model to include bioconcentration at every trophic level.

The bioaccumulation of  $\mathrm{Hg}^{2+}$  is much less studied and not incorporated in any of the above-mentioned models. This is because  $\mathrm{Hg}^{2+}$  is much less toxic than  $\mathrm{MMHg}^+$  and therefore comparably understudied. While data is limited, this raises the speculative question if the link between the bioaccumulation  $\mathrm{Hg}^{2+}$  and  $\mathrm{MMHg}^+$  is not underestimated as  $\mathrm{Hg}^{2+}$  and  $\mathrm{MMHg}^+$  are in active equilibrium in the water.

The ECOSMO-MERCY coupled system, which is used by Bieser et al. (2023) is the only coupled model that models the bioaccumulation of  $Hg^{2+}$  and  $MMHg^{+}$  at higher trophic levels such as fish, while incorporating bioconcentration at every trophic level. The version used by Amptmeijer et al. (2025) expands on this by adding a higher-trophic-level fish. Because of this, the ECOSMO-MERCY coupled system, as described by Amptmeijer et al. (2025) is used in this analysis.

**Implementation**

**Line 260**

The implementation of bioaccumulation is discussed and validated in more detail in Amptmeijer et al. (2025), but the core equations are discussed here as well for clarity. The increase in bioconcentrated pollutant (Hg2+ or MMHg+) per day for a functional group is calculated based on the biomass concentration of the group, the uptake rate, and the concentration of the pollutant, while it is reduced with a rate that is the sum of the release rate of the pollutant and the loss of biomass from group g, from both biological loss (respiration and mortality) and predation. The change in pollutant p due to bioaccumulation can then be calculated using the following equation:

$$\frac{dC_{g,p}^{BC}}{dt} = b_g \cdot C_p^{env} \cdot r_{g,p}^{bc} - C_{g,p}^{BC} \cdot (r_{g,p}^{rel} + r_g^{bl} + \sum_{z=1}^{n_z} r_{z,g}^{pred})$$
 (13)

```
\begin{split} C_{g,p}^{BC} &= \text{Bioconcentrated pollutant } p \text{ in group } g \text{ [ng Hg m}^{-3}] \\ b_g &= \text{Biomass of functional group } g \text{ [mgC m}^{-3}] \\ C_p^{env} &= \text{Environmental concentration of pollutant } p \text{ [ng Hg m}^{-3}] \\ r_{g,p}^{bc} &= \text{Bioconcentration rate for group } g \text{ and pollutant } p \text{ [m}^3 \text{ mgC}^{-1} \text{ d}^{-1}] \\ r_{g,p}^{rel} &= \text{Release rate of pollutant } p \text{ from group } g \text{ [d}^{-1}] \\ r_{z,g}^{gl} &= \text{Biological loss rate for group } g \text{ (mortality, respiration) [d}^{-1}] \\ r_{z,g}^{pred} &= \text{Predation rate by predator } z \text{ on group } g \text{ [d}^{-1}] \\ n_z &= \text{Number of consumer groups feeding on group } g \\ z &= \text{Index for consumer groups (predators) of } g \\ t &= \text{Time [d]} \end{split}
```

While the change in pollutant p due to biomagnification is also dependent on the predation and concentration of pollutants from both bioconcentration and biomagnification in the prey. Additionally, pollutant p is released via the turnover rate rather than the release rate, as is the case for bioconcentration. The change in pollutant p due to biomagnification can then be calculated as follows:

$$\frac{dC_{g,p}^{BM}}{dt} = \sum_{s=1}^{n_s} (r_{g,s}^{pred} \cdot a_{s,p} \cdot (C_{s,p}^{BC} + C_{s,p}^{BM})) - C_{g,p}^{BM} \cdot (r_{p,g}^{to} + r_g^{bl} + \sum_{z=1}^{n_z} r_{z,g}^{pred})$$
(14)

 $C_{q,p}^{BM} = \text{Pollutant } p \text{ concentration in group } g \text{ from biomagnification [ng Hg m}^{-3}]$

 $n_s$  = Number of prey groups consumed by g

s =Index for prey functional groups of g

 $r_{q,s}^{pred}$  = Predation rate of group g on prey group s [d-1]

 $a_{s,p} =$ Assimilation efficiency of pollutant p from prey s [unitless]

 $C_{s,p}^{BC}$  = Pollutant p concentration in group s from bioconcentration [ng Hg m-3]

 $C_{s,p}^{BM}$  = Pollutant p concentration in group s from biomagnification [ng Hg m-3]

 $r_{p,q}^{to}$  = Turnover rate of pollutant p in group g [d-1]

 $r_g^{bl}$  = Biological loss rate for group g [d-1]

 $r_{z,q}^{pred}$  = Predation rate of predator z on group g [d-1]

So the total concentration of pollutant P in ng Hg m-3 is:

$$C_{(g,p)} = C_{(g,p)}^{BC} + C_{(g,p)}^{BM}$$
(15)

Since this tracks the pollutants per volume of water, the total bioaccumulation per biomass in ng Hg mgC-1 is then calculated as

$$C_{(g,p)}^{bg} = \frac{C_{(g,p)}}{b_a} \tag{16}$$

This is then converted to the bioaccumulation per dry weight based on an assumed ratio of carbon to dry weight of 0.2 for diatoms, 0.33 for flagellates and cyanobacteria, and 0.5 for zooplankton and fish based on Walve and Larsson (1999) and Sicko-Goad et al. (1984).

**Implementation**

**Line 330**

**Sensitivity analyses**

In order to further investigate how bioconcentration in consumers affects bioaccumulation of MMHg+, we performed a sensitivity analysis on the key drivers: the bioconcentration rate of consumers and the bioaccumulation rate of producers. To this extent, two sensitivity studies are performed. In the first sensitivity study, the bioconcentration rate in all consumers is multiplied by a scaling factor that is between 0.2 and 2.0 with 0.2 intervals. The effect of this on the bioaccumulation in fish 2 for the Gotland Deep is shown to visualize the impact. Then the relative contribution of bioconcentration in consumers on the bioaccumulation of MMHg+ in fish 2 is shown for all three setups. For the second sensitivity study, the same approach is used but the bioconcentration rate of producers is multiplied by a scaling factor.

**Implementation**

**Line 335**

**Results and discussion**

The model output is shown in Table 1. The % bioconcentrated is calculated as Bioconcentrated (%) =  $\frac{\text{Bioconcentrated}}{\text{Bioaccumulated}}$  and the difference (%) is calculated as Difference (%) =  $\frac{\text{Scenario}}{\text{Default}}$ . The thick values in the difference category indicate when the scenario causes a change larger than 10%. The values are based on the last 10 years of the simulation and the top 20m of the water column, to create an average value that we can compare between the setups.

**Bioaccumulation of Hg2+**

The effect of  $\mathrm{Hg^{2+}}$  bioaccumulation on the bioaccumulation of MMHg+ is shown in Table 1. All results are derived from model simulations. To quantify the influence of consumer level bioconcentration and bioaccumulation of  $\mathrm{Hg^{2+}}$  on MMHg+ bioaccumulation, the model was run under scenarios with and without bioaccumulation of  $\mathrm{Hg^{2+}}$  and with and without consumer-level bioconcentration of MMHg+. The differences are low between 1% and -6%. This is statistically evaluated, and the results are shown in Table 2. Wilcoxon's signed rank test shows that bioaccumulation of  $\mathrm{Hg^{2+}}$  has no significant impact on the bioaccumulation of MMHg+ (p = 0.67). Furthermore, the Bayesian t-test shows that the data are 2.9 times more likely under the null hypothesis of no effect than under the alternative hypothesis. This shows that  $\mathrm{Hg^{2+}}$  bioaccumulation does not have a significant effect on MMHg+ bioaccumulation (BF10=0.40).

**Bioaccumulation of MMHg+**

The MMHg+ bioaccumulation for all biota functional groups in the different setups and scenarios and the percentage of bioaccumulated MMHg+ originating from bioconcentration are shown in Table 1. These results show that the relative contribution of bioconcentration on the MMHg+ content is low in microzooplankton (4–6%) while it is higher in mesozooplankton (5–10%) higher in fish 1 (13–22%), while lower in fish 2 (8–14%) and higher in macrobenthos (14–25%). The relative contribution of direct bioconcentration on

the MMHg+ bioaccumulation in zooplankton, especially microzooplankton, is lower than in higher trophic levels of animals. In our model, this occurs because of the extremely high turnover rate of zooplankton. This "grow fast, die young" approach results in less MMHg+ bioconcentration with higher relative contributions due to feeding caused by the high feeding rate of zooplankton.

In longer-lived fish, we see higher contributions of bioconcentration. Although these contributions are higher, they align with the experiments of (Wang & Wong, 2003) and the observations of 15% by Hall et al. (1997). Both fish 1 and fish 2 have the same bioconcentration and release rates, so it is in line with expectations that the relative contribution of direct bioconcentration in fish 2 is lower than in fish 1 since it gets more MMHg+ from its higher trophic level diet.

There is a great difference in the importance of bioconcentration of MMHg+ in macrobenthos between the Southern and Northern North Sea. This difference is especially notable in the direct bioconcentration in macrobenthos, which is 25% of the total bioaccumulated MMHg+ in the Northern North Sea and only 14% in the Southern North Sea. This difference is caused by the low intake of MMHg+ from food by macrobenthos in the Northern North Sea. Since the water column is stratified during spring and summer, macrobenthos cannot directly feed on the phyto- and zooplankton bloom. Because of this, they are dependent on sinking detritus. The detritus has a lower MMHg+ content than living material and consequently, the MMHg+ intake in Northern North Sea macrobenthos is lower, and thus the relative importance of bioconcentration is higher.

**Implementation**

Line 378

**Sensitivity analyses**

**Sensitivity of the consumer bioconcentration rate**

The results of the first sensitivity study, in which the bioconcentration rate of consumers is altered, are shown in Fig. 3. Figure 3a illustrates that the MMHg+ contribution from bioconcentration in consumers is linearly related to the consumer bioconcentration rate scaling factor. Thus, in bioaccumulation modeling, altering the bioconcentration rate by half or double yields the same relative effect on fish 2's MMHg+ content from direct bioconcentration. Based on Table 1, we can see that in the Gotland Deep, the difference between the simulation with and without consumer bioconcentration is 0.0183 ng Hg mgC-1. This means that parameterizing a bioconcentration rate double the real rate would result in a 0.0183 ng Hg mgC-1 overestimation of MMHg+ bioaccumulation in fish 2, while selecting bioconcentration rates half the true values would result in a reduction of 0.00915 ng Hg mgC-1. However, the relative contribution of bioconcentration to total MMHg+ bioaccumulation follows a non-linear pattern, as shown in Fig. 3b. This non-linearity occurs because the total MMHg+ in fish 2 is influenced by both bioconcentration in consumers and bioconcentration in producers. When the consumer bioconcentration scaling factor is 0, bioconcentration in consumers makes no contribution to fish 2's MMHg+ levels. Conversely, this contribution can never reach 100% because bioconcentration in producers and consequent biomagnification from lower trophic levels always contributes to the total MMHg+ burden in fish. In the same way as in the results shown in ??, the relative importance of bioconcentration is consequently highest in the Northern North Sea, followed by the Southern North Sea and lowest in the Gotland Deep.

**Sensitivy of the producer bioconcentration rate**

The results of the second sensitivity study are shown in Fig. 2. Here, rather than the consumer bioconcentration rate, the producers' bioconcentration rates are multiplied by a scaling factor. Again, the effect of this scaling on the bioaccumulation in all trophic levels is visualised in Fig. 2a, and the effect of this scaling on the relative importance of consumer bioconcentration on MMHg+ bioaccumulation is shown in Fig. 2b. If the bioconcentration scaling factor is 0, there is still MMHg+ bioaccumulation in fish 2, both from direct bioconcentration and from bioconcentration in consumers and consequent biomagnification. The increase in fish 2 MMHg+ per step of 0.2 in the scaling factor is  $0.0083 \pm 0.00030$  ng Hg mg-1. The relative contribution of consumer bioconcentration on MMHg+ bioaccumulation is shown in Fig. 2b. An important note here is that while we scaled the bioconcentration factor of producers and consumers, MMHg+ can also be bioaccumulated via the partitioning to DOM detritus and consequent biomagnification. This is especially important in the Northern North Sea. In the seasonally stratified water column, macrobenthos cannot feed directly off the phytoplankton bloom; thus, the dying and sinking of particles is an important flux that is consumed by the benthos. Benthos, in turn, is an important food source for fish 2. So scaling the producer bioconcentration rate has less effect in the Northern North Sea. In the Gotland Deep, the opposite is true; because the deep water is anoxic, there is no macrobenthos in the model. This means that the entire ecosystem is pelagic and detritus is less important than direct consumption of the phytoplankton bloom.

**Seasonality of the difference in MMHg+ bioaccumulation**

The seasonality of the difference in MMHg+ bioaccumulation caused the bioaccumulation of Hg2+ and the bioconcentration of MMHg+ in consumers is shown in Fig. ??. For each calendar day (January 1st, January 2nd, etc.), the modeled daily values from each of the last 10 years of the simulations were averaged. The resulting time series represents an annual cycle of average daily conditions. From the producers' functional groups, only the diatoms are shown as the reaction is not group-specific but rather caused by changes in dissolved Hg2+ and MMHg+ which means the difference caused for all phytoplankton groups was the same. This shows that, while the scale depends on the setup, there are interactions that consistently occur. In low trophic levels, such as phytoplankton and microzooplankton, the bioaccumulation of Hg2+ causes a seasonal response in the MMHg+ bioaccumulation in phytoplankton, which is consequently observable in low trophic level biota such as microzooplankton. While this reduction in MMHg+ would compound into higher trophic levels, its effects in higher trophic level animals dwarf in comparison to the difference caused by incorporating the bioconcentration of MMHg+ in consumers, and it does not cause a difference larger than 3% in either fish 1 or fish 2 in any of the setups.

Figure 1: a) show the effect of the bioaccumulation of MMHg+ per trophic level in the Gotland Deep. This shows an increase  $0.0036 \pm 0.00010$  ng Hg mg-1 in fish 2 MMHg+ bioaccumulation for every 0.2 step increase in the consumer bioconcentration scaling factor. 3b) shows the percentage difference due to bioaccumulation with different consumer bioconcentration scaling factors in all setups. GD refers to the Gotland Deep, NNS to the Northern North Sea and SNS to the Southern North Sea. When the consumers bioconcentration scaling factor is 0, the percentage difference due to bioconcentration is 0%. As this increase the percentage increases. The relationship between the consumers bioconcentration factor and the percentage difference due to consumer bioconcentration is plotted assuming an saturating exponential relationship.

Figure 2: a) illustrates the influence of scaling the producers bioconcentration rate of MMHg $^+$  on the MMHg $^+$  bioaccumulation at each trophic level in the Gotland Deep. This shows an increase of  $0.0084 \pm 0.00032$  ng Hg mg $^{-1}$  in fish 2 MMHg $^+$  with every 0.2 increase in the producers scaling factor. 2b) highlights the relative significance of bioconcentration across all setups. The relationship between producer bioconcentration scaling factor and the percentage difference in MMHg $^+$  bioaccumulation in fish 2 due to consumer bioconcentration is plotted using an expenontial decay function. This shows that in all cases the percentage difference is high when the producer bioconcentration factor is 0, and that this percentages decreases with an increasing scaling factor.

Figure 3: a) show the effect of the bioaccumulation of MMHg+ per trophic level in the Gotland Deep. This shows an increase  $0.0036 \pm 0.00010$  ng Hg mg-1 in fish 2 MMHg+ bioaccumulation for every 0.2 step increase in the consumer bioconcentration scaling factor. 3b) shows the percentage difference due to bioaccumulation with different consumer bioconcentration scaling factors in all setups. GD refers to the Gotland Deep, NNS to the Northern North Sea and SNS to the Southern North Sea. When the consumers bioconcentration scaling factor is 0, the percentage difference due to bioconcentration is 0%. As this increase the percentage increases. The relationship between the consumers bioconcentration factor and the percentage difference due to consumer bioconcentration is plotted assuming an saturating exponential relationship.

**Implementation**

**Line 589**

**Uncertainty of the conclusion**

The results of our model represent just one possible outcome based on a regional setup representing the North and Baltic Seas, and the importance of bioconcentration can vary greatly depending on the bioconcentration factors of all species in the trophic chain. We can assess the expected range of importance of consumer level bioconcentration by developing theoretical maximum and minimum values based on observational studies. We can estimate that direct bioconcentration in zooplankton may account for up to 50%, based on Lee and Fisher (2017), and similarly for mid-trophic level fish, based on Wang and Wong (2003).

We can use this to estimate the maximum expected contribution of consumer-level bioconcentration on bioaccumulation by making two assumptions: (1) bioconcentration in both copepods and fish lies between 0 and 50% and is equal across all trophic levels, and (2) the food chain is linear, meaning that trophic level 3 feeds exclusively on trophic level 2, which feeds exclusively on trophic level 1. Under these assumptions, we can estimate the percentage of MMHg+ in the diet of a given trophic level that originated from bioconcentration in primary producers as:

$$PBC\%_n = (1 - BC)^{n-1} \times 100\%$$
(17)

where:

- PBC%n is the percentage of MMHg+ in the diet of trophic level n that originates from bioconcentration at the primary producer level,
- BC is the fraction (0–1) of MMHg+ at each trophic level originating from bioconcentration.

Although this is a simplification, it illustrates that a high bioconcentration estimate of 50% results in only 12.5% of MMHg+ in the diet of a trophic level 4 fish originating from

bioconcentration in primary producers, meaning that 87.5% originates from consumer-level processes. Even a low estimate of 10% still results in 27.1% of MMHg+ in the diet of the same high-trophic-level fish originating from consumer-level bioconcentration. The degree to which this interaction contributes to overall bioaccumulation depends on numerous additional factors that are not yet fully understood, including the size distribution of phytoplankton at the base of the food web, the trophic structure, consumer metabolic and respiration rates, and the assimilation efficiency of MMHg+ from the diet. This complexity makes it difficult, if not impossible, to provide a definitive estimate of the importance of consumer-level bioconcentration and the uncertainty of the interaction. However, based on the bioconcentration rates provided in the current literature, we conclude that this process plays a key role in the bioaccumulation of MMHg+ in higher trophic levels.

**Reviewer Comment**

"Line 40-50. The authors mentioned the "bioconcentration", "bioaccumulation" and "biomagnification". For instance, the statement "bioconcentration is the most important step in bioaccumulation" lacks a clear distinction from biomagnification, risking confusion for readers unfamiliar with the terminology. What's the differences between bioaccumulation and biomagnification? In the subsequent manuscript, these two words were also used in confusion. The authors should explain and clarify them. "

**Author Response**

This distinction should indeed have been clarified better, especially since it is essential for understanding the paper. We have expanded the text starting at line 41 as described above to ensure clarity of the terms used and describe them in detail. Then we added a segment at line 260 to describe in details the equations used in this study to model bioaccumulation, including the different equations for bioaccumulationg originating from bioconcentration of biomagnification.

**Reviewer Comment**

"Line 79-85. The second hypothesis is confused. This hypothesis lacks evidence and references, making it appears speculative."

**Author Response**

I agree that the second hypothesis lacked references. It is mostly supported by the work of Wu et al. (2019), so we rewrote it as below to increase clarity.

**Implementation**

**Line 140**

The majority of MMHg+ present in higher trophic levels is derived from their dietary intake (Lavoie et al., 2013). It is often assumed that MMHg+ bioconcentration is not crucial for its bioaccumulation at higher trophic levels based on results such as those presented by Schartup et al. (2018), it is, for example, omitted from several Hg cycling and bioaccumulation models such as the model presented by Rosati et al. (2022), or not incorporated into higher trophic levels, as is the case in the model presented by Li et al. (2022). However, this assumption overlooks that bioconcentration occurs at all levels of the trophic chain. For example, if microzooplankton and mesozooplankton acquire 5% of MMHg+ through bioconcentration, mesozooplankton will have 5% less MMHg+ from its diet, which consists of microzooplankton, and another 5% less due to the absence of bioconcentration, leading to a total reduction of 10%. The second hypothesis is that MMHg+ bioconcentration in consumers significantly elevates MMHg+ levels at higher trophic levels. This concept has been previously suggested and studied by Wu et al. (2019). Their research found that the BCF in fish spans 3 to 7 orders of magnitude and greatly differs across studied sites; yet, they did find a strong correlation between BCF and MMHg+ concentration in fish. Thus, we are not the first to suggest that direct water uptake is a significant factor in MMHg+ bioaccumulation; rather, this study extends this understanding by quantifying the role of bioconcentration in all consumers on MMHg+ bioaccumulation in fish at higher trophic levels.

**Reviewer Comment**

"Line 140: As mentioned, "Quantifying the importance of the bioaccumulation of Hg2+ and bioconcentration of MMHg+ in consumers on MMHg+ bioaccumulation". The authors should clarify whether prior models ignored multi-trophic bioconcentration, and then highlight the novelty of this work."

**Author Response**

The section described above from line 108 has been updated to give a fair summary of previously published models and how they incorporated bioaccumulation and why the modelling framework used is selected.

**Reviewer Comment**

"Line 226 Table 1. How to calculate the bioaccumulation and bioconcentration difference (%)?

**Author Response**

We added additional formula on line 339 and in the caption of Table 1.

**Implementation**

**Line 339**

The % bioconcentrated is calculated as Bioconcentrated (%) =  $\frac{\text{Bioconcentrated}}{\text{Bioaccumulated}} * 100\%$  and the difference (%) is calculated as Difference (%) =  $\frac{\text{Scenario}}{\text{Default}} * 100$ . The values in red in the difference category indicate when the scenario causes a change larger than 10%. The values are based on the last 10 years of the simulation and the top 20m of the water column, to create an average value that we can compare between the setups.

**Implementation**

**Caption of Table 1**

The bioaccumulated MMHg+, the percentage of bioaccumulated MMHg+ that originates from bioconcentraton, and the bioaccumulated MMHg+ in the scenario without bioaccumulation of Hg2+ and the bioconcentration of MMHg+ in consumers and the difference to the default scenario. The % bioconcentrated is calculated as Bioconcentrated (%) =  $\frac{\text{Bioconcentrated}}{\text{Bioaccumulated}} * 100$  and the difference (%) is calculated as Difference (%) =  $(1 - \frac{\text{Scenario}}{\text{Default}}) * 100$ .

**Reviewer Comment**

Line 316 "15% per trophic level". It is recommended to supplement sensitivity analyses.

We supplemented the paper with a sensitivity study as described above. It is introduced in line 330 in the Methods sections and the results are shown in line 378 in the result section.

**Reviewer Comment**

The introduction mentions "MMHg+is a topic of serious concern because MMHg+is a dangerous neurotoxin that can bioaccumulate to levels. that are dangerous for human consumption in fish that are often consumed as seafood." Could this phrase be a little more concise?

**Author Response**

We have changed that sentence as described below:

**Implementation**

**Line 15**

The element mercury (Hg) is presently included in the World Health Organization's list of the 10 substances of greatest concern (WHO, 2020). This is due to the capability of Hg to be methylated to form monomethyl mercury (MMHg+), a potent neurotoxin generated by microbial methylation of inorganic Hg. MMHg+ biomagnifies within aquatic food webs, accumulating in predatory fish to concentrations that can impair human neurological development upon consumption.

**Reviewer Comment**

The introduction mentions "Since DMHg is susceptible to photodegradation, we can assume that it plays an important role in the coastal water investigated in this study, until better observational studies confirm or correct this assumption." There are problems with logic

**Author Response**

That is indeed incorrect, and we corrected that as described below:

**Implementation**

**Line 29**

However, given the rapid photodegradation of DMHg in natural water and that it is generally not assumed to bioaccumulate, DMHg is assumed not to significantly bioaccumulate in biota in the coastal area investigated in this study (Morel et al., 1998; West et al., 2022).

**Reviewer Comment**

The introduction mentions "the bioconcentration process is complicated and recent studies show that the bioconcentration of MMHg+is influenced by cell-dependent factors, such as the thickness of the phytosphere, while this is not the case for Hg2+." How does the thickness of the phytosphere affect the bioconcentration of MMHg+? Why is Hg2+ not affected by this?

There is a lot of uncertainty about these processes and they are somewhat out of the scope of this modeling paper. Therefore I will enhance the unclear sentence with the expanded sentence belows, in order to create more clarity about this interaction while not distracting from the focus of the paper

**Implementation**

**Line 56**

However, the bioconcentration process is controlled by a variety of factors, and recent studies show that the bioconcentration of  $\mathrm{Hg}^{2+}$  is constant when normalized for cell density, while the uptake of  $\mathrm{MMHg}^+$  is affected by changes in cell density and biomass. This suggests that  $\mathrm{MMHg}^+$  uptake is influenced by cell-dependent factors, such as the thickness of the phycosphere and the availability of transmembrane channels, while this is not the case for  $\mathrm{Hg}^{2+}$  (Garcia-Arevalo et al., 2024).

**Reviewer Comment**

The references cited in the manuscript are somewhat outdated and may benefit from incorporating more recent studies to ensure the relevance and accuracy of the presented information.

**Author Response**

The sections described above (Especially the implementation at line 108) have updated references. Additionally some updated references are added below.

**Implementation**

**Line 19**

For example, it is estimated that the consumption of MeHg contaminated seafood contributed to 61,800 premature deaths and caused economic damage of up to 2.87 trillion USD (Chen et al., 2025). This issue is expected to become even more significant as antrhopgenic Hg emissions are projected to increase in the near future (Maria Brocza et al., 2024).

**Line 34**

Additionally, it has been shown by Tesán-Onrubia et al. (2023) that plankton communities in the southern Mediteranean Sea have lower MMHg+ concentration than plankton in the northern Mediterranean Sea, they linked this to changes in environmental conditions affecting bioconcentration.

**Reviewer Comment**

The paper is very detailed about the basic knowledge and the content of the preliminary research, but the description of the later model establishment and the data obtained from the model is relatively brief, which can be further supplemented. For example, "The contribution of bioconcentration in zooplankton of 3.97-10.07%..... and the contribution of bioconcentration in fish between 8.14-21.82%...."

The results and discussion are rewritten as is shown above starting from ( the segment from line 335) to be more preciese about the results. Additionally, in the 376 the seasonality from the modeled data is also plotted and evaluated to further expand on evaluating the model data.

**Reviewer Comment**

- Section 2.2: Are there any equations for the parameterization schemes in the bioaccumulation of Hg in the model? Model equations are important for understanding the critical processes of the substance. Meanwhile, what are the critical parameters and coefficients for the critical processes in the model? The details of this model are not clarified in the method.
- Significantly, model performance should be evaluated against observations, which are deficient in this study. The literature Amptmeijer et al. (2025) is very important for this study. However, we cannot access to the paper because it is in preparation.

**Author Response**

My apologies. The publication of a key paper for this model was delayed. These details of this model are referenced in Amptmeijer et al. 2025, which is currently available with https://doi.org/10.5194/egusphere-2025-1486 I believe this is the main concern of the 3 points above. Here, the key equations and validation of both carbon fluxes and bioaccumulation are discussed in more detail. I addition to this paper now beign available I would discuss in more detail exactly which equations we use. This is described at the beginning of this document, and starting from line 38 to clarify the terminology and 260 to clarify the equations in the updated manuscript. Additionally, the below described summary of the performance of the model from Amptmeijer et al. (2025) is added on line 302.

**Implementation**

**Line 302**

**Performance of the GOTM-ECOSMO-MERCY model**

The model performance is discussed in more detail in Amptmeijer et al. (2025), but the key metrics are summarized below. The model is generally consistent with observational data and the previously validated 3D ECOSMO E2E model in terms of biomass. Minor exceptions are that the Chlorophyll-a concentration in the Gotland Deep matches the Northern instead of the Central Baltic Sea, and that the fish biomass in the Gotland Deep is overestimated by 7% compared to Thurow (1997). The model also predicts tHg content in phytoplankton, zooplankton, and fish 1 accurately, and the MMHg+ bioaccumulation in fish corresponds well with trophic interactions. A deviation is seen in the trophic level fish 2, which has a trophic level between 3.5 and 3.7 in the model, below the expected level for Atlantic Cod (between 4.0 and 4.2). Nonetheless, this level remains high, making fish 2 representative of high-trophic-level animals. The MMHg+ bioaccumulation in fish 2 is consistent with the observed bioaccumulation for its trophic level. Thus with the above-discussed minor exceptions, the model simulates biomass, Hg speciation, and bioaccumulation in line with observations.

**Reviewer Comment**

The setup of the two scenarios is a sample. In my opinion, sensitivity analysis for critical parameters or uncertainty analysis of the results is needed.

I agree that the results of the paper where too limited and underexplored. As described above, in order support the manuscript a sensitivity study was performed. The text is at the top of the document, but the results of the sensitivity and study start form line 378 in the manuscript.

**Reviewer Comment**

For the results and discussion, the illustrations are concise, and I cannot gain much indepth discussion and thinking.

**Author Response**

In addition to the sensitivy study we also performed a seasonality analyses on the model data (at line 386 in the manuscript) and expanded the discussion starting form line 589 as is described above. I hope this provides a deeper layer of desired depth to the manuscript.

**Reviewer Comment**

This study employed different models to identify the contributions of bioconcentration and biomagnification to Hg and MeHg bioaccumulation. The primary concern is that the description of the model applied and the data sources lack clarity.

**Author Response**

Before going into the specific comments, I would answer your general feedback. The model does not use field data and is solely based on a model. The core model cited is presented in Amptmeijer et al. (2025). Which is discussed and evaluated here in more detail: https://doi.org/10.5194/egusphere-2025-1486 The model presented by Amptmeijer et al. (2025) is then used to run with and without bioaccumulation of Hg2+ and consumer-level bioconcentration of MMHg+ to estimate the importance of these interactions. As such, the core message of the paper is aimed at showing the importance of these interactions on the outcome of the model, which shows that these interactions should be included in MMHg+ bioaccumulation models. The method section is expanded, notably from line 41 to better describe the used terminology and from line 260 to show in detail the equations used in this study. I hope thas makes clarifies the model descriptions.

**Reviewer Comment**

Lines 6-9: This sentence is too long and not clear to me

**Implementation**

**Line 6**

In this study, we use a fully coupled 1D water column Hg bioaccumulation model to quantify how total bioaccumulation of Hg2+ and uptake of MMHg+ from the water (bioconcentration) in consumers affects the bioaccumulation of MMHg+ in high-trophic-level fish.

**Reviewer Comment**

Line 110: Descriptions about the Modeled region in the Introduction section is weird. I suggest moving it to the MM section.

**Author Response**

Moved this to the Methods section at **line** 183

**Reviewer Comment**

The model section is not clear to me. How to divide bioconcentration and biomagnification. Is there any data collected from in-lab measurements?

**Reviewer Comment**

Table 1: What is the source of the data provided in this table?

I would address these comments together as they addres the same issue. This study is purely model-based. Of course, previously published data collected from in-lab measurements are used to estimate the bioconcentration rates and assimilation efficiency, which drive the processes in our model. This is discussed in detail in the referenced model paper Amptmeijer and Bieser (2025). Which is available here in preprint: https://doi.org/10.5194/egusphere-2025-1486. I agree that we underexplained the difference between bioconcentration and biomagnification and how this is done in our model. I hope the expansion of the methods section starting at line 41 in the manuscript and described above addresses this concern. Additionally I would suggest to add the below statement at the beginning of the results section at line 210 to make sure there is no ambiguity about the source of the data.

**Implementation**

Line 343

**Results and discussion**

The model output is shown in Table 1. All results are derived from model simulations. To quantify the influence of consumer-level bioconcentration and bioaccumulation of  $\mathrm{Hg^{2+}}$  on  $\mathrm{MMHg^{+}}$  bioaccumulation, the model was run under scenarios with and without bioaccumulation of  $\mathrm{Hg^{2+}}$  and with and without consumer-level bioconcentration of  $\mathrm{MMHg^{+}}$ . The % bioconcentrated is calculated as Bioconcentrated (%) =  $\frac{\mathrm{Bioconcentrated}}{\mathrm{Bioaccumulated}} * 100\%$  and the difference (%) is calculated as Difference (%) =  $\frac{\mathrm{Scenario}}{\mathrm{Default}}$ . The thick values in the difference category indicate when the scenario causes a change larger than 10%. The values are based on the last 10 years of the simulation and the top 20m of the water column, to create an average value that we can compare between the setups.

---

## Author Response (AR2)

**1 Answer to reviewer 1**

**Reviewer Comment**

Line 40: in vivo Hg speciation? or in vivo methylation/demethylation. Here, it appears that the authors are primarily focused on in vivo methylation. Although there is currently no conclusive evidence supporting in vivo methylation in marine organisms, in vivo demethylation has been widely documented. Therefore, the use of the term "in vivo Hg speciation" should be approached with caution or avoided.

**Author Response**

Thank you very much for reviewing the paper a second time. I fully agree with this comment. Based on some of the modern work, I find that there is evidence indicating more complex in vivo Hg chemistry. This is, however, very early stage research and distracts from the main aim of the paper, which is focused on understanding the more established pathway of bioaccumulation. I also realize that in the discussion, I do not evaluate this, as it is out of the main scope of the paper. As such, I have removed the section about in vivo Hg speciation and all of the references to this. I believe this would make an interesting topic for a dedicated modelling study but agree it should be discussed without being expanded upon in this manuscript.

**Reviewer Comment**

2. Conclusion section: no need to repeat the hypotheses here

**Author Response**

I removed the first part of the conclusion where I restated the hypothesis. Now the conclusion starts directly with the results.

**2 Answer to reviewer 2**

**Reviewer Comment**

Thanks for carefully addressing my comments, and the manuscript has been improved greatly. However, I have a very serious concern when I access to the nearly contemporaneous paper from the authors, which is also under review (https://doi.org/10.5194/egusphere-2025-1486). Why the authors separate the bioconcentration process from the model in that paper? If the bioconcentration process very important, the model performance from that paper should not achieve the best condition in my opinion. Considering the similar study regions and models, especially the nearly same titles, I have doubts about these two articles being online at the same time. What's the necessities, differences, and improvements?

**Author Response**

Thank you for revieuwing the work a second time and pointing this out, and I hope my suggestion below clarifies this concern. The difference is that the other paper is focused on analyzing the role of the ecosystem in Hg cycling and the feedback mechanisms between bioaccumulation and Hg cycling. Examples are how bioaccumulation affects sedimentation, evaporation, the Hg budget, and Hg transport. This paper, on the other hand, is focused on analyzing the design choices that set this model apart from other Hg bioaccumulation models. Notably, the inclusion of bioconcentration of MMHg at every trophic level and the bioaccumulation of Hg2+. The honest reason why this is presented as a second paper rather than included in the other paper is two-fold. First, as mentioned before, the aim of the paper is different, and the paper is already 48 pages, not including the supplement. Initially, we wanted to make one combined paper but decided to make 1 paper for readers that were interested in how the ecosystem affects Hg cycling and another paper where we critically evaluate model design choices. We decided that presenting the messages that bioconcentration of Hg2+ might not be essential for Hg bioaccumulation modeling but that consumer level bioconcentration is, would be better in a more streamlined stand-alone paper mostly focused to help further Hg bioaccumulation model development. Because of this, I would suggest adding the following at the end of the introduction:

**Implementation**

**Line 122**

To clarify, the base case used in this study is identical to the base case of the 1D model presented in Amptmeijer et al., 2025, where the bioaccumulation of both  $\mathrm{Hg^{2+}}$  and  $\mathrm{MMHg^{+}}$  is represented through biomagnification and bioconcentration across all simulated trophic levels. The difference lies in the direction and scope of the analyses. In Amptmeijer et al., 2025, the model was evaluated for both carbon stocks and  $\mathrm{Hg^{2+}}$  and  $\mathrm{MMHg^{+}}$  to demonstrate that it reasonably reproduces the removal of  $\mathrm{Hg}$  from the water column via bioaccumulation. This evaluation enabled an assessment of the feedback mechanism of bioaccumulation on  $\mathrm{Hg}$  cycling and the overall  $\mathrm{Hg}$  budget. In contrast, the present study focuses on a sensitivity analysis of two specific model design choices made in Amptmeijer et al., 2025: the incorporation of consumer-level bioconcentration of  $\mathrm{MMHg^{+}}$  and the bioaccumulation of  $\mathrm{Hg^{2+}}$ .

---

## Author Response (AR3)

Note to the editor.

I read over the manuscript once more and realised there was something that could be clarified better. As such I added the statement to line 379 in the results and discussion section:

"The percentage differences between the base case and the alternative setups are also shown in Table 1. Notably, the change in MeHg concentrations is substantially larger in the scenario without consumer-level bioconcentration. The greatest difference between the base case and the setup without consumer-level bioconcentration occurs in fish 2, with increases ranging from 28% to 49%, while this effect is smaller in lower-trophic level biota. Interestingly, in the Southern North Sea, a reduction in the bioaccumulation of MMHg \*in primary producers is observed in both the scenario without Hg² + bioaccumulation (3–6%) and the scenario without consumer-level MMHg \* bioconcentration (3–7%). This reduction is likely linked to the removal of bioaccumulated Hg from the water column by macrobenthos feeding on pelagic food sources and, consequently, it is not present in the two deep-water setups. Nevertheless, the overall effect on MMHg \* bioaccumulation remains limited between -2% and 2%.

I also added the DOI of the code with the below statement to line 561

"The model code is publicly available on Zenodo (DOI: 10.5281/zenodo.17372353) under the GPL 3.0 License."

I hope adding this minor clarification is fine. Alternatively, I can upload a version where these parts are not present.